# Design and optimisation of dendrimer-conjugated Bcl-2/x$_L$ inhibitor, AZD0466, with improved therapeutic index for cancer therapy

Claire M. Patterson[1,8], Srividya B. Balachander[2,8], Iain Grant[3,8], Petar Pop-Damkov[2], Brian Kelly [4], William McCoull [5], Jeremy Parker [1], Michael Giannis [4], Kathryn J. Hill [3], Francis D. Gibbons [2], Edward J. Hennessy [2], Paul Kemmitt [5], Alexander R. Harmer [6], Sonya Gales [6], Stuart Purbrick [6], Sean Redmond [7], Matthew Skinner [6], Lorraine Graham [1], J. Paul Secrist [2], Alwin G. Schuller [2], Shenghua Wen[2], Ammar Adam[2], Corinne Reimer[2], Justin Cidado [2], Martin Wild [5], Eric Gangl[2], Stephen E. Fawell[2], Jamal Saeh[2], Barry R. Davies [5], David J. Owen [4] & Marianne B. Ashford [1]✉

Dual Bcl-2/Bcl-x$_L$ inhibitors are expected to deliver therapeutic benefit in many haematological and solid malignancies, however, their use is limited by tolerability issues. AZD4320, a potent dual Bcl-2/Bcl-x$_L$ inhibitor, has shown good efficacy however had dose limiting cardiovascular toxicity in preclinical species, coupled with challenging physicochemical properties, which prevented its clinical development. Here, we describe the design and development of AZD0466, a drug-dendrimer conjugate, where AZD4320 is chemically conjugated to a PEGylated poly-lysine dendrimer. Mathematical modelling was employed to determine the optimal release rate of the drug from the dendrimer for maximal therapeutic index in terms of preclinical anti-tumour efficacy and cardiovascular tolerability. The optimised candidate is shown to be efficacious and better tolerated in preclinical models compared with AZD4320 alone. The AZD4320-dendrimer conjugate (AZD0466) identified, through mathematical modelling, has resulted in an improved therapeutic index and thus enabled progression of this promising dual Bcl-2/Bcl-x$_L$ inhibitor into clinical development.

[1] Pharmaceutical Sciences, R&D, AstraZeneca, Macclesfield SK10 2NA, UK. [2] Oncology R&D, AstraZeneca, Boston, MA 02451, USA. [3] Pharmaceutical Technology and Development, Operations, AstraZeneca, Macclesfield SK10 2NA, UK. [4] Starpharma, 4-6 Southampton Crescent, Abbotsford, VIC 3067, Australia. [5] Oncology R&D, AstraZeneca, Cambridge CB4 0FZ, UK. [6] Clinical Pharmacology & Safety Sciences, R&D, AstraZeneca, Cambridge CB4 0FZ, UK. [7] Clinical Pharmacology & Safety Sciences, R&D, AstraZeneca, Boston, MA 02451, USA. [8]These authors contributed equally: Claire M. Patterson, Srividya B. Balachander, Iain Grant. ✉email: Marianne.Ashford@astrazeneca.com

The induction of apoptosis in tumour cells represents a promising approach to the treatment of cancer. The B cell lymphoma 2 (Bcl-2) protein family regulates the mitochondrial (intrinsic) apoptosis pathway[1]. In tumour cells, upregulation of anti-apoptotic Bcl-2 proteins, such as Bcl-2, Bcl-$x_L$, Mcl-1 and Bcl-w prevent apoptosis and can lead to chemoresistance[2]. Compounds that interact with the Bcl-2 family of proteins have been widely pursued as potential anticancer agents. While encouraging anti-tumour activity in clinical trials has led to the approval of the Bcl-2 inhibitor, venetoclax, the therapeutic utility of dual Bcl-2/Bcl-$x_L$ inhibitors such as navitoclax has been limited by accompanying toxicities. Dual Bcl-2/Bcl-$x_L$ inhibitors are expected to provide greater therapeutic benefit in many haematological cancers solid tumours (where Bcl-$x_L$ overexpression predominates)[3,4]. However, Bcl-$x_L$ also has a role in normal tissue especially maintaining platelet survival[5]. In fact, thrombocytopenia is one of the main dose-limiting toxicities of the oral Bcl-2/Bcl-$x_L$ inhibitor, navitoclax[6,7].

AstraZeneca's dual Bcl-2/Bcl-$x_L$ inhibitor, AZD4320, was intended as an intravenously administered drug with an intermittent dosing schedule, with the aim of achieving rapid Bcl-2/Bcl-$x_L$ inhibition in tumour cells whilst enabling recovery from on-target thrombocytopenia between dosing[8]. However, whilst the goal of achieving a potent dual Bcl-2/Bcl-$x_L$ inhibitor with activity across a broad range of tumour types with manageable thrombocytopenia risk was achieved dose limiting cardiovascular toxicity was observed during preclinical development thus AZD4320 could not be progressed to the clinic. These dose-limiting adverse signs were concomitant with several cardiovascular changes: dose-dependent decreases in QRS amplitude, QRS duration & PR intervals, blood pressure, left ventricular pressure and contractility (Supplementary Fig. 1a, b). These findings occurred below predicted efficacious exposures.

Additionally, AZD4320 has very low solubility (<1 μg/ml in aqueous buffers at physiological pH) and plasma protein binding was high across pre-clinical species and man with unbound fraction being <0.003%. Preclinical efficacy was achieved using hydroxypropyl-β-cyclodextrin as a carrier to solubilise AZD4320 and enhance tumour exposure. Unusually this efficacy and tumour exposure was shown to be formulation dependent (Supplementary Fig. 2) which is thought to be due to a combination of a particularly high degree of both cyclodextrin and protein binding, as well as the short half-life of the drug in plasma). The lack of therapeutic index and delivery concerns prompted the exploration of drug delivery opportunities for this promising candidate drug to enable its clinical development.

Nanomedicines have been extensively investigated to improve the therapeutic index of cytotoxic drugs with prior market approval[9,10]. Typically, this has been with the aim of improving the toxicological or pharmacokinetic profile of the cytotoxic[11] but in solid tumour indications, nanomedicine delivery may additionally improve efficacy through enhanced tumour accumulation[12,13] or in haematological tumours by extended circulation time. In contrast, very few studies have investigated nanomedicine delivery of molecularly targeted small molecule drugs. A nanoparticle formulation of the Aurora kinase B inhibitor, AZD2811, is a key example of this class[14–16].

Dendrimers are branched polymers that are built up of concentric rings of monomer, also known as generations. Drug dendrimer conjugates offer many advantages as a nanomedicine[17,18]; their size can be determined via generation and they are typically smaller than many nanoparticles (~7-15 nm), they are water soluble, have near-monomodal molecular weight distributions, reproducible synthesis steps and have large numbers of surface groups available for conjugation. The linker chemistry through which drug conjugation is achieved can be tailored to enable controlled release or site-specific release of the active moiety; a factor critical for improving therapeutic index[19].

The DEP® dendrimer platform, developed by Starpharma, is based on a fifth generation PEGylated poly-L-lysine dendrimer. DEP® docetaxel is in Phase 2 clinical trials (EudraCT Number: 2016-000877-19) and is the most advanced DEP® clinical programme. In the clinic, DEP® docetaxel has been reported to cause less neutropenia and avoid the excipient-related toxicities associated with Taxotere®[20]. The synthetic chemistry underlying the manufacture of DEP® dendrimers enables their precise and reproducible synthesis. This is a key consideration in the selection of nanomedicine technologies for clinical development and commercialisation as chemistry, manufacturing and control remains underappreciated[11].

A diverse range of different modelling techniques have been instrumental in improving the understanding of nanomaterial behaviour in biological systems in cancer nanomedicine[21]. Physiologically based pharmacokinetic (PB-PK) mathematical modelling is routinely used and critical to understanding the absorption, distribution, metabolism and elimination of drugs and needs to be adapted and used more in nanomedicine design[22].

Here we have used mathematical modelling to design a dendrimer conjugate of AZD4320 with an optimal release rate for favourable therapeutic index in terms of pre-clinical efficacy and cardiovascular tolerability. We have used an iterative approach of design, synthesis, modelling and in vivo testing to optimise the structure of the AZD4320-dendrimer conjugate. The size of such a conjugate is sufficiently large to preclude rapid renal excretion, the PEGylation has been demonstrated to reduce recognition by the reticuloendothelial system (RES)[23,24] and the size small enough to enable good tumour penetration. The combined desired effect is a reduced plasma $C_{max}$, prolonged circulation time and maximal potential for accumulation of the dendrimers within tumours. Release of the active moiety is via hydrolytic cleavage of the chemical linker and thus the rate can be tailored through linker design. Initially we synthesised three drug-dendrimer conjugates with a range of AZD4320 release half-lives to enable tolerability and efficacy to be investigated. In parallel, using experimental data from these three initial conjugates, a mathematical model was developed to describe the in vivo behaviour of the drug-dendrimer conjugates. The model was subsequently used to evaluate the impact of release rate on in vivo performance (namely efficacy and cardiovascular tolerability) and to optimise the design of the drug-dendrimer conjugate for further evaluation and development.

## Results

**Synthesis and characterisation of AZD4320-dendrimer conjugates with different linker chemistries.** Three drug-dendrimer conjugates (SPL-8931, SPL-8932 and SPL-8933) were initially synthesised where the active moiety, AZD4320, was chemically conjugated to the free lysines on a PEGylated fifth-generation (G5) poly-L-lysine dendrimer through either a glutarate, thiol-diglycolate and diglycolate linker (Fig. 1, Supplementary Figs. 3–6) on its primary alcohol. The aim was to provide a wide range of AZD4320 release rates for investigation of efficacy and tolerability.

Characterisation data for representative batches of the drug-dendrimer conjugates is presented in Supplementary Table 1. Typically, molecular weight was in the region of ~105 kDa and loading was 24–30% w/w (25–32 molecules) AZD4320, where the maximum theoretical achievable loading is 32 AZD4320 molecules. Solubility of the conjugates in aqueous buffer is high (>100 mg/ml). This represents a great improvement on solubility of AZD4320 (<1 μg/ml) and facilitates formulation and delivery

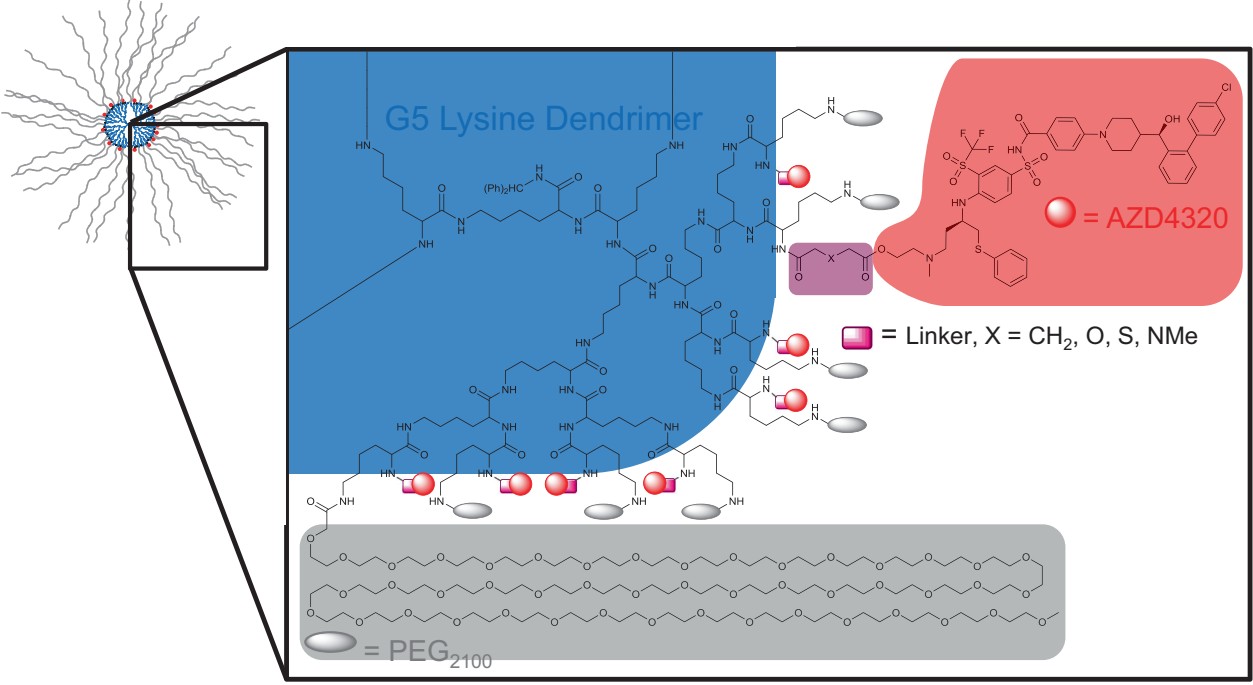

**Fig. 1 Structure of initial AZD4320-dendrimer conjugates.** Overall schematic of PEG$_{2100}$ containing AZD4320-dendrimer conjugates showing structure and chemical structure of highlighted quadrant. Conjugation sites and synthetic route allow for a maximum of 32 conjugated PEG molecules (grey ovals) and 32 AZD4320 molecules (red dots) per dendrimer (blue circle). Variability in linker (pink rectangle): X = CH$_2$ for SPL-8931, X = S for SPL-8932 and X = O for SPL-8933.

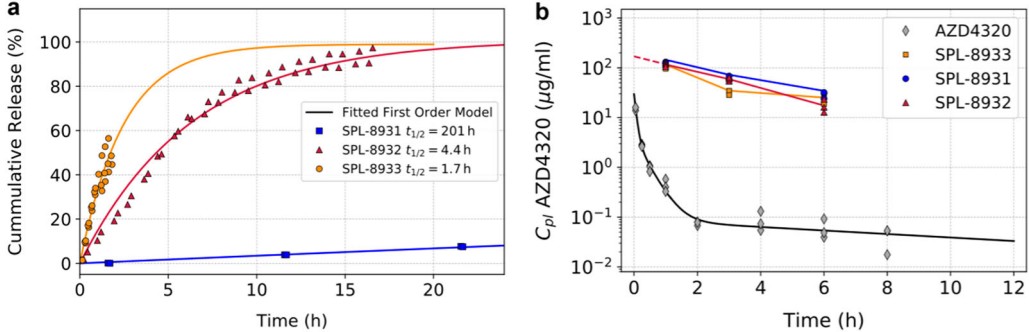

**Fig. 2 Cumulative in vitro release of AZD4320 from AZD4320-dendrimer conjugates and mouse pharmacokinetics of AZD4320 and AZD4320 delivered via dendrimer conjugates. a** Cumulative release of AZD4320 from SPL-8931 (blue squares), SPL-8932 (red triangles) and SPL-8933 (orange circles) in phosphate buffered saline at pH 7.4, 37 °C and fit to first order release model ($n = 2/3$). **b** Plasma pharmacokinetics following intravenous dose of AZD4320 (grey diamonds) and AZD4320 delivered via SPL-8931 (blue circles), SPL-8932 (red triangles) and SPL-8933 (orange squares) drug-dendrimer conjugates in C.B-17 SCID mouse at 10 mg/kg AZD4320 dose or equivalent ($n = 3$).

of AZD4320-dendrimer conjugates by the intravenous route. The drug-dendrimer conjugates typically have measured hydrodynamic diameters of ~10 nm (PDI <0.2).

The release of AZD4320 from SPL-8931, SPL-8932 and SPL-8933 in phosphate-buffered saline (PBS), pH 7.4, at 37 °C followed first order kinetics and gave release half-lives of 201, 4.4 and 1.7 h respectively (Fig. 2a). This broad range enabled the impact of AZD4320 release rate on tolerability, pharmacodynamic effect and efficacy to be investigated.

**Pharmacokinetics of AZD4320 and initial drug-dendrimer conjugates in plasma.** The pharmacokinetics of AZD4320 and its dendrimer conjugates were evaluated by administering a bolus intravenous (IV) injection of either AZD4320 or SPL-8931, SPL-8932 and SPL-8933 to C.B-17 SCID mice. The AZD4320 concentration in the plasma drops, showing a very rapid alpha phase

to less than 20% of the initial concentration within 10 min (Fig. 2b). Conversely the three AZD4320-dendrimer conjugates show a much more prolonged circulation time typical of a dendrimer with PEGylated surface designed to minimise uptake by the RES organs and one whose size is too big to be rapidly cleared by the kidney[23]. An initial volume of distribution of 59 ml/kg was calculated which is similar to plasma volume in the mouse[25]. The similar behaviour of the drug-dendrimer conjugates with very different release half-lives shows that the release rate plays a relatively small part in the decline of the total drug concentrations in plasma. This suggests that RES uptake is a predominant factor in clearance and is similar across the drug-dendrimer conjugates.

**Pharmacodynamics and efficacy of AZD4320 and initial drug-dendrimer conjugates in a mouse xenograft model.** In vivo activity and tumour kinetics of the drug-dendrimer conjugates

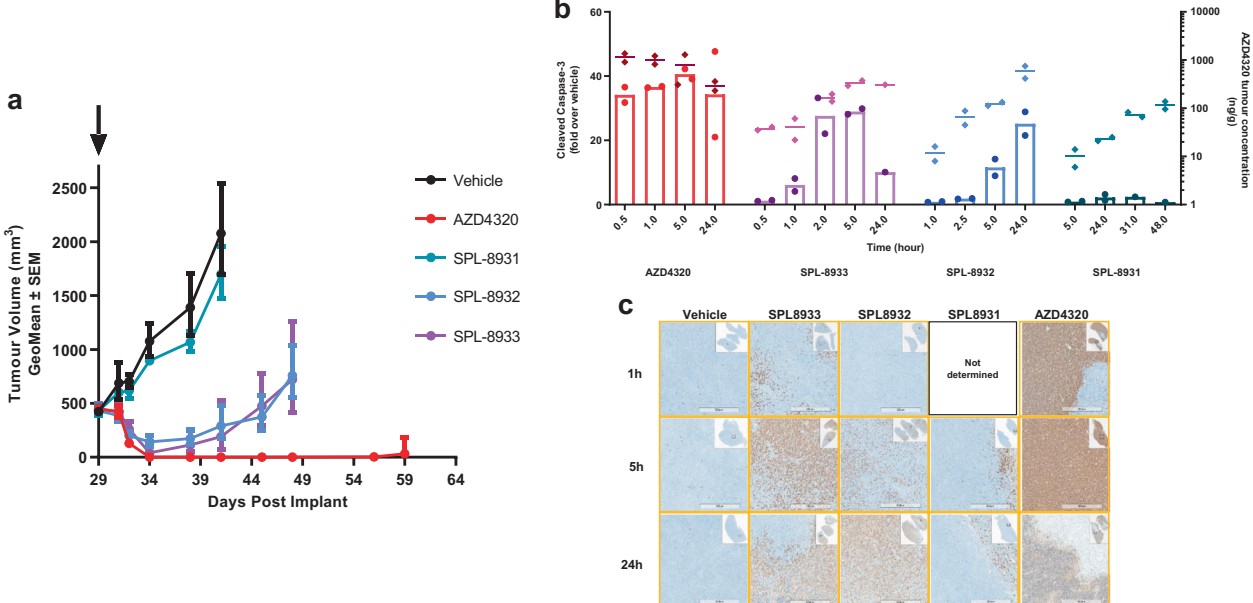

**Fig. 3 In vivo PD, efficacy and tumour pharmacokinetics of AZD4320 and initial drug-dendrimer conjugates. a** Tumour growth inhibition of RS4;11 xenograft of AZD4320 (red) and initial AZD4320-dendrimer conjugates SPL-8931 (cyan), SPL-8932 (blue) and SPL-8933 (purple) (10 mg/kg or equivalent dose). Data in the graphs are represented as mean tumour volumes ±SEM (*n* = 6 per group). **b** Cleaved caspase in RS4;11 tumours measured by MSD ELISA (left axis) and tumour pharmacokinetics of AZD4320 (right axis) from single dose of AZD4320 (10 mg/kg) and dendrimer conjugates (equivalent to 10 mg/kg AZD4320). Data in the graphs are represented as mean (*n* = 2 per time point). **c** Tumours taken at 1, 5, and 24 h post dose were stained for CC3 by IHC (No sample taken from SPL-8931 treated animals at 1 h due to the long AZD4320 release half-life). Scale bar 500 μm.

were investigated by administering an IV injection of either AZD4320 or SPL-8931, SPL-8932 and SPL-8933 to mice bearing RS4;11 tumours. The IV dose of AZD4320 demonstrated strong anti-tumour activity resulting in complete regression of the RS4;11 tumours. The two dendrimer conjugates with the faster release half-lives, SPL-8933 and SPL-8932, initially produced tumour regression with similar kinetics to AZD4320, however the tumours eventually re-grew and only showed tumour growth inhibition (TGI) of 65 and 67% respectively. No anti-tumour activity was observed after dosing 10 mg/kg SPL-8931, the dendrimer conjugate with the longest release half-life (Fig. 3a).

To understand the differential response, caspase induction in the RS4;11 tumours was quantified and correlated to the extent of caspase induction observed in vivo to efficacy. AZD4320 induced rapid activation of cleaved caspase-3 (CC3) within 0.5 h, reached maximal activation of 40-fold by 5 h that was sustained for 24 h. Kinetics of caspase activation by the AZD4320-dendrimer conjugates tracked with their release half-lives (Fig. 3b). SPL-8933 with the shortest release half-life rapidly induced a 27-fold caspase-activation at 2 h, while a more delayed response was seen with SPL-8932 with onset at 5 h and maximal activation of 25-fold at 24 h. CC3 induction with SPL-8931 was very low (2.5-fold) and occurred at 24 h. Neither SPL-8932 and SPL-8933 achieved the sustained maximal caspase levels of AZD4320 and hence complete regression was not maintained at the doses tested. The caspase analysis is mirrored by the immunohistochemistry (IHC) data for cleaved caspase-3 in these tumour samples showing robust cleaved caspase induction with AZD4320 and to a lesser extent with SPL-8932 and SPL-8933 (Fig. 3c). The concentration of AZD4320 in the tumour was measured. As shown in right *y*-axis of Fig. 3b, AZD4320 rapidly reached maximum levels in the tumour within 0.5 h of dosing. In contrast SPL8933 provided a maximum released AZD4320 tumour concentration by 5 h while SPL8932 achieved a highest concentration at 24 h. The released concentration of AZD4320 from SPL-8931 was negligible. This released AZD4320 concentration correlates with both the caspase

activation levels observed in the tumours and the differential efficacy observed.

## Nonclinical tolerability and cardiovascular effects of AZD4320 and initial drug-dendrimer conjugates (SPL-8931 and SPL-8932).

Two of these initial drug-dendrimer conjugates with different release half-lives, SPL-8931 and SPL-8932 were assessed in a rat telemetered model in order to determine the effects on QRS amplitude and compared to those of a similar dose of AZD4320. SPL-8933 was omitted from the study due to its very rapid release half-life. AZD4320 clearly shows a marked and prolonged reduction in QRS amplitude (Fig. 4a) and was only tolerated in 6 of the 8 rats evaluated. A 2-minute IV infusion of SPL-8932 was associated with some reduction in QRS amplitude (Fig. 4b) but to a lesser extent than the equivalent dose of AZD4320 as well as a decrease in platelet count (a measure of Bcl-x$_L$ activity) (Supplementary Fig. 7; Supplementary Table 2). SPL-8931, the construct with longest release half-life, had no cardiovascular effects in telemetered rats at this dose (Fig. 4b), but also had less effects on platelet count (Supplementary Fig. 7). Both SPL-8931 and SPL-8932 were well tolerated (Supplementary Table 3). These findings indicated that a release half-life more than 5 h was desirable to avoid cardiovascular effects.

## Mathematical modelling to guide design of further drug-dendrimer conjugates.

A mathematical model was constructed to describe aspects of the in vivo disposition of AZD4320 dosed as drug-dendrimer conjugates (Fig. 5a). This model assumes that when a drug dendrimer conjugate is injected into the bloodstream there are three options; either the drug-dendrimer conjugate is cleared from the circulation intact (this is expected to be predominantly via the organs of the RES[23]), is extravasated intact into tumour tissue or drug is released from the conjugate into the circulation. In addition, the model also assumes that hydrolysis release rate at pH 7.4 in buffer is the same as in plasma at 37 °C

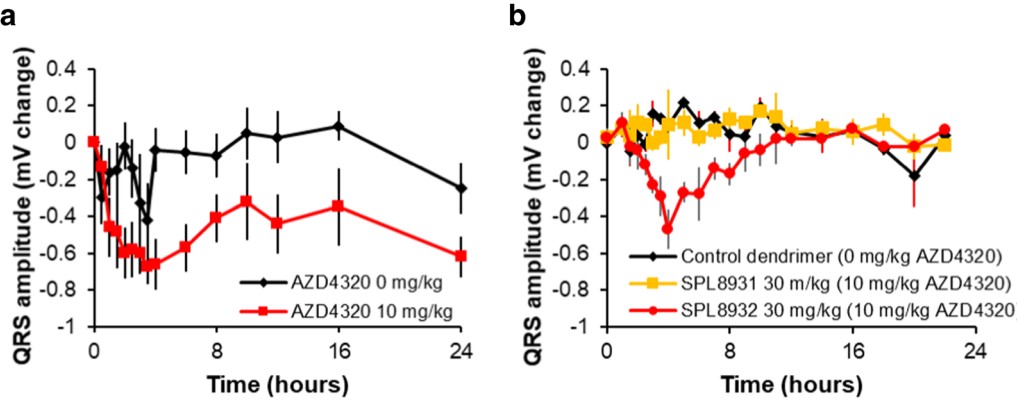

**Fig. 4 Cardiovascular effects of AZD4320 and AZD4320-dendrimer conjugates in rat.** Time course of changes in QRS amplitude in telemetered rats following single dose IV administration at $t = 0$ h of **a** vehicle (0 mg/kg) AZD4320, (black diamonds) and AZD4320 (red squares), $n = 6\text{-}8$ per group. **b** Control dendrimer (black diamonds) SPL-8931 (orange squares) and SPL-8932 (red circles), $n = 3$ per group. Data are group mean ± SEM.

(Supplementary Fig. 8). The objective was to obtain mathematical expressions to predict the impact of linker hydrolysis rate on the concentration of released AZD4320 in tumours and in plasma, thereby guiding optimal linker design. These were then used to determine the release half-life required for optimal balance between high maximum concentrations of released AZD4320 in the tumour (to drive efficacy) and sufficiently low plasma $C_{max}$ (linked to tolerability) for a given dose. Table 1 gives a summary of equations derived for the concentrations of total and released AZD4320 in both plasma and in tumour tissue.

Using the respective hydrolysis rate constants ($k_h$) calculated from the release half-life profiles of the dendrimer conjugates and fitting Eq. (1) to the data for the plasma concentration profile of SPL 8932, the mid releasing dendrimer, enabled a value for the rate constant for reticuloendothelial system (RES) uptake ($k_{res}$) to be estimated (0.21 h$^{-1}$). This is depicted in Fig. 5b and shows a good fit to the experimentally derived data. The $k_{res}$ value from SPL8932 was assumed to apply to all the dendrimers evaluated in the study given that alternative linker chemistry represents a relatively small change to these very large molecules and thus RES uptake is expected to be similar. To determine $k_{ext}$, the dendrimer extravasation constant and $k_{tu}$, the constant for return of released AZD4320 from tumour tissue to systemic circulation and to fully parameterise the model, the best fit to Eq. (3) was determined by simultaneous optimisation of $k_{ext}$ and $k_{tu}$ to concentration-time profiles of released AZD4320 in tumour tissue for the initial three dendrimer constructs (SPL-8933, SPL-8932 and SPL-8931, Fig. 5c). Best fit values for the dendrimer extravasation rate constant ($k_{ext}$) and rate of return of released AZD4320 from tumour to plasma ($k_{tu}$) were 3.8 x 10$^{-4}$ h$^{-1}$ and 0.053 h$^{-1}$ respectively. It is acknowledged that $k_{ext}$ depends on tumour type and size, but in this model, the optimal value of AZD4320 release rate ($k_h$) is independent of the absolute value of $k_{ext}$. The parameters used in the modelling derived from this initial set of experimental data (pharmacokinetics and efficacy) are shown in Table 2.

Figure 5d shows there is a release half-life, corresponding to approximately 5 h, which is predicted to give the highest maximum concentration of released AZD4320 in the tumour tissue for a given dose however longer releasing half-lives will only result in a slight drop in tumour $C_{max}$ per unit dose. In contrast, the model predicts that plasma $C_{max}$ per unit dose drops much more rapidly with increasing release half-life than the predicted tumour $C_{max}$ (Fig. 5e). Thus, for longer releasing half-lives, the ratio of $C_{max}$ in the tumour to $C_{max}$ in the plasma increases implying that the therapeutic index should continue to increase with increasing release half-life. However, the total dose

required to get an equivalent maximum concentration in the tumour will also need to be increased for slower releasing dendrimers which would ultimately lead to impractically large doses of the drug-dendrimer conjugate to achieve activity. To mitigate this the impact of dose must be factored into the overall optimisation process.

For a given release half-life, the predicted total dose required for an equivalent tumour $C_{max}$ of released drug is inversely proportional to the predicted tumour $C_{max}$ for that same half-life. Therefore, to take dose into account, an overall optimisation index was defined where the predicted tumour $C_{max}$ to plasma $C_{max}$ ratio is effectively divided the required dose. As a result, we end up with the square of the predicted tumour $C_{max}$ per unit dose over the predicted plasma $C_{max}$ per unit dose. Figure 5f shows how this optimisation index changes with the release half-life and illustrates the predicted optimal hydrolysis half-life for AZD4320, taking efficacy, tolerability and dose into account, is in the approximate range of 20 to 30 h.

This modelling guided a further round of synthesis aimed at achieving release rates in the optimal range through alternative linker chemistries. SPL-8974, SPL-8976 and SPL-8977 were synthesised and found to have in vitro release half-lives of 12, 7 and 25.5 h respectively. The mechansim of release was hydrolysis. The plasma profiles of AZD4320 (conjugated and released drug) from these three additional drug-dendrimer conjugates are very similar to those of the initial three drug-dendrimer conjugates, this is shown in Supplementary Fig. 9. These new conjugates were dosed in the RS4;11 mouse xenograft model and good alignment between predicted and observed tumour concentrations of released AZD4320 (Fig. 5g) and total concentration of AZD4320 (Supplementary Fig. 10) was observed . Fig. 5h shows good agreement between predicted and observed tumour concentration both for total and released AZD4320 for SPL8977. The correlation between predicted and measured released AZD4320 tumour concentrations from all six drug-dendrimer conjugates is shown in Supplementary Fig. 11 and supports the predictive power of the model. Taking the prediction from Fig. 5f, SPL-8977 (hereafter referred to as AZD0466), with the AZD4320 release half-life of 25.5 h within the optimal range, was selected for further in vivo efficacy and tolerability assessment. The structure and characterisation are shown in Supplementary Fig. 12 and Supplementary Table 3.

**Optimised AZD4320-dendrimer conjugate, AZD0466 exhibits potent efficacy in mouse xenograft model.** The efficacy of

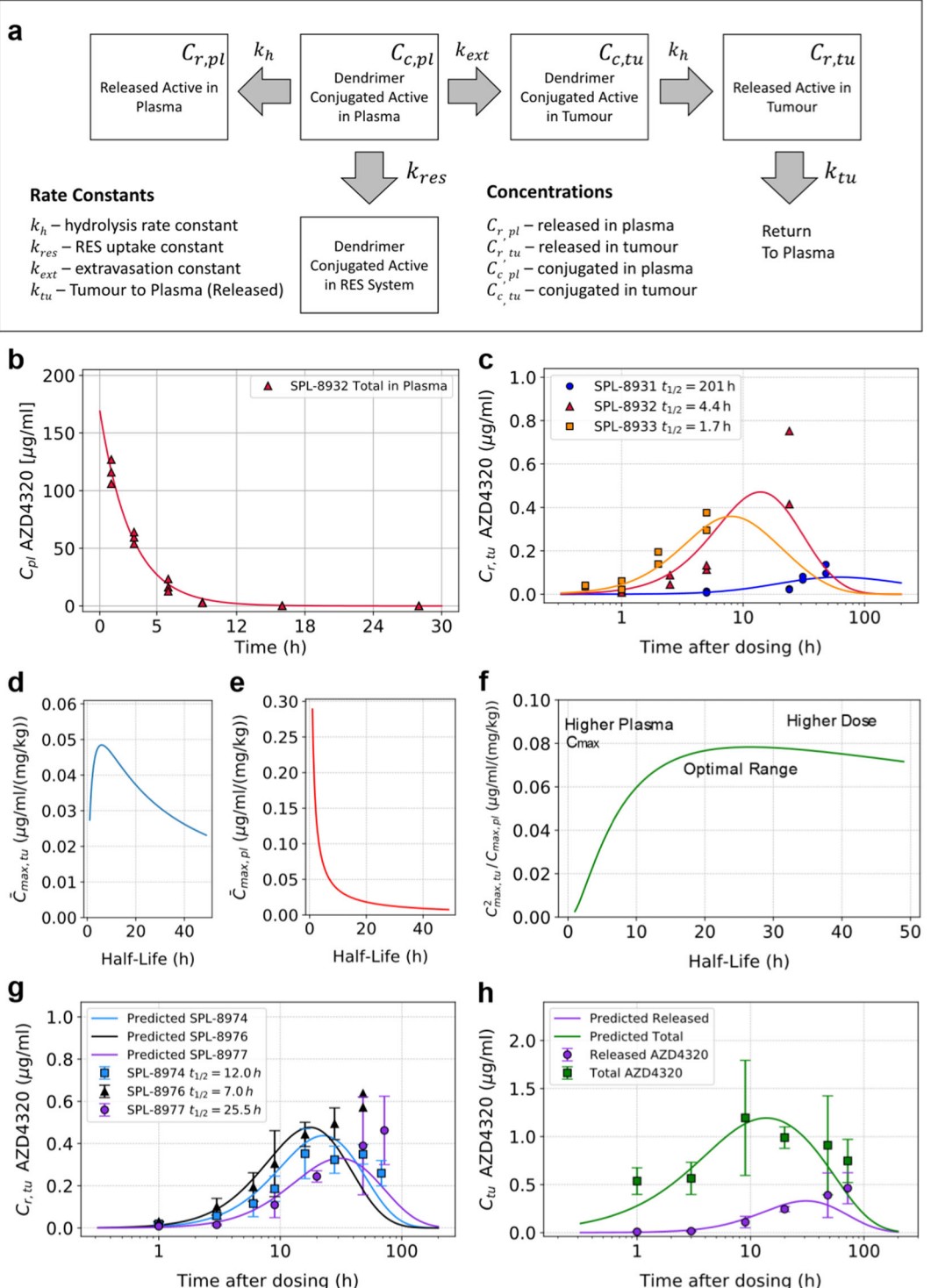

**Fig. 5 In silico model for AZD4320-dendrimer conjugates. a** Schematic illustration of the model describing the impact of linker hydrolysis rate on AZD4320 concentrations in tumour and plasma. **b** Model fitting to total AZD4320 concentrations in plasma for SPL-8932 following IV dose to mouse giving a $k_{res}$ value of 0.21 h$^{-1}$ ($n=3$). **c** Fitted model vs observed released tumour concentration ($n = 2$) of AZD4320 for SPL-8931 (blue circles), SPL-8932 (red triangles) and SPL-8933 (orange squares), (all 10 mg/kg AZD4320 equivalent). **d, e** Simulated dependence of released AZD4320 tumour and plasma $C_{max}$ on release half-life (per unit dose). **f** Optimisation Index (Therapeutic Index per unit dose) as a function of release half-life. **g** Predicted (lines) vs observed (symbols, mean ± SD on $n = 3$) released AZD4320 in tumour for SPL-8974 (blue squares), SPL-8976 (black triangles) and SPL-8977 (purple circles), (all 10 mg/kg AZD4320 equivalent). **h** Predicted (lines) vs observed (symbols, mean ± SD on $n = 3$ animals) tumour released (purple circles) and total concentrations (green squares) of AZD4320 for SPL-8977 (dosed at 10 mg/kg AZD4320 equivalent). The total concentrations in tumour for all dendrimers are included in Supplementary Fig. 10.

**Table 1 Key equations for concentrations of AZD4320 in tumour tissue and plasma.**

| | |
|---|---|
| 1. Concentration of Conjugated API in Plasma | $C_{c,pl}(t) = \frac{D}{V_{pl}} \exp\left(-(k_{res} + k_h)t\right)$ |
| 2. Concentration of Conjugated API in Tumour Tissue | $C_{c,tu}(t) = \frac{D}{V_{tu}} \frac{k_{ext}}{k_{res}} \exp(-k_h t)\left(1 - \exp(-k_{res} t)\right)$ |
| 3. Concentration of Released API in Tumour Tissue | $C_{r,tu} = \alpha \frac{D}{V_{tu}} \left( \begin{array}{c} \beta \exp(-k_{tu} t) + \gamma \exp\left(-(k_h + k_{res})t\right) \\ -\delta \exp(-k_h t) \end{array} \right)$ |
| 4. Concentration of Released API in Plasma (approx for $t < 2$ hr) | $C_{r,pl}(t) \approx \omega \frac{D}{V_c} \left( \exp\left(-(k_{el} + k_{12} + k_{13})t\right) - \exp\left(-(k_h + k_{res})t\right) \right)$ |

Summary of equations derived for the concentrations of total and released AZD4320 in both plasma and in tumour tissue for a dose ($D$) as a function of time ($t$).
Note: $D$ is the dose given as mass of AZD4320.
For Eqs. (3) and (4) the constant terms $\alpha$, $\beta$, $\gamma$, $\delta$, $\omega$ and $k_{el}$ are defined as, $\alpha = \frac{k_{ext} k_h}{k_{res}}, \beta = \frac{k_{res}}{(k_h - k_{tu} + k_{res})(k_h - k_{tu})}, \gamma = \frac{1}{k_h - k_{tu} + k_{res}}, \delta = \frac{1}{(k_h - k_{tu})}, \omega = \frac{k_h}{k_h + k_{res} - (k_{el} + k_{12} + k_{13})}, k_{el} = \frac{CL}{V_c},$
$k_h$ is related to the release half-life($t_{1/2}$), by $k_h = \ln(2)/t_{1/2}$.

**Table 2 Key parameters derived from the experimental data for the model.**

| Parameter description | Symbol | Value | Unit |
|---|---|---|---|
| Dendrimer Parameters | | | |
| Dendrimer Extravasation Constant (first order) | $k_{ext}$ | 0.00038 | h$^{-1}$ |
| Dendrimer RES Uptake Constant (first order) | $k_{res}$ | 0.21 | h$^{-1}$ |
| Release Rate Constant (first order) – SPL-8931 | $k_h(t_{1/2})$ | 0.00345 (201 h) | h$^{-1}$ |
| Release Rate Constant (first order) – SPL-8932 | $k_h(t_{1/2})$ | 0.158 (4.4 h) | h$^{-1}$ |
| Release Rate Constant (first order) – SPL-8933 | $k_h(t_{1/2})$ | 0.408 (1.7 h) | h$^{-1}$ |
| AZD4320 Parameters | | | |
| Constant for return of released AZD4320 from tumour tissue to systemic circulation (first order) | $k_{tu}$ | 0.053 | h$^{-1}$ |
| Total Plasma Clearance | $CL$ | 0.0296 | L/h |
| Central Compartment Volume | $V_c$ | 0.0042 | L |
| Compartmental Micro Constants | $k_{12}$ | 2.31 | h$^{-1}$ |
| | $k_{21}$ | 0.108 | h$^{-1}$ |
| | $k_{13}$ | 3.92 | h$^{-1}$ |
| | $k_{31}$ | 4.36 | h$^{-1}$ |
| In Vivo Parameters (used in model) | | | |
| Mouse Plasma Volume | $V_{pl}$ | 59.2 | mL/kg |
| Tumour Volume at Time of Dosing | $V_{tu}$ | 300 | mm$^3$ |

AZD0466 was evaluated at three different doses in the Bcl-2 dependent RS4;11 mouse xenograft model. A weekly intravenous dose of AZD0466 at 3, 10 and 30 mg/kg AZD4320-equivalent dose resulted in tumour regression in all 6 mice within each dose group. Complete tumour regression was achieved with 10 mg/kg and 30 mg/kg doses during the treatment period (Fig. 6a) with 2/6 animals remaining tumour free for up to 6 weeks at the highest dose. A single administration of AZD0466 resulted in a dose and time-dependent induction of cleaved caspase 3 (Fig. 6b). The cleaved caspase induction levels were consistent with the concentrations and kinetics of released AZD4320 measured in the tumour (Fig. 6c). All treatments were well tolerated with no obvious signs of adverse events and minimal body weight loss (Supplementary Fig. 13). In a separate study the efficacy of AZD0466 to SPL-8932 at 20 mg/kg AZD4320 equivalent dose and to 10 mg/kg AZD4320 was compared. The magnitude of in vivo efficacy of AZD0466 was comparable to that of SPL-8932 and AZD4320 (Supplementary Fig. 14) as predicted from the modelling.

**Nonclinical tolerability and cardiovascular effects of AZD0466.** Next, we assessed if AZD0466 provided better therapeutic margin than SPL-8932 by measuring QRS amplitude in a rat telemetry study. AZD0466 at 30 mg/kg dose had no effect on QRS amplitude in a rat telemetry study (Fig. 7a); this is in contrast to that of AZD4320 itself and the faster releasing conjugate SPL 8932 at similar doses of AZD4320 (Fig. 4a, b). Thus, AZD0466 with its slower release rate, behaves as predicted by the modelling and has a minimal effect on QRS amplitude.

AZD0466 was also subjected to a fuller investigation in a dog telemetry study, where a single dose of 60 mg/kg AZD0466 was not associated with any adverse clinical signs. This is very different to the effect seen at much lower doses of AZD4320 (Supplementary Fig. 1, Fig. 7b). In conscious telemetered dogs, a 1-hour infusion of AZD0466 at 30 and 60 mg/kg caused a dose-dependent decrease in QRS amplitude that returned to baseline, with no obvious related impact on other cardiovascular parameters (Fig. 7b). A small increase in diastolic BP, a transient and delayed-onset small increase in heart rate, a decrease in dP/dt+ (an index of cardiac contractility) and a decrease in PR interval were also observed (Fig. 7c). The effect on QRS amplitude was much less marked with AZD0466 compared to a 0.5-hour infusion of AZD4320, for instance 1 mg/kg of AZD4320 (Supplementary Fig. 1) gave a greater decrease in QRS amplitude than 60 mg/kg AZD0466 (equivalent to 20 mg/kg AZD4320). 10 mg/kg AZD0466 was established as the NOEL for effects on the cardiovascular system in dog and was well tolerated. The equivalent AZD4320 dose (3 mg/kg) exceeds the maximum tolerated dose of AZD4320 in dogs. AZD0466 also caused a reversible and dose-dependent decrease in platelet count that recovers to normal levels before the next weekly dose, indicating a pharmacological effect on Bcl-x$_L$ activity at all dose levels (Fig. 7d).

**AZD0466 efficacy in a disseminated model and in combination with other agents.** Disseminated xenograft models are considered to be more representative of many haematological malignancies in which disease burden is widely disseminated throughout the body[26]. In order to evaluate the anti-tumour activity of AZD0466 in a disseminated setting a luciferase-tagged Ri-1-DLBCL tumour model was employed and the effect of AZD0466 treatment on disease burden was assessed via whole animal imaging and measurement of bioluminescence. A once weekly intravenous administration of AZD0466 demonstrated dose-dependent anti-tumour activity in this model. (Fig. 8a, b). These data indicate that the dendrimer-conjugate AZD0466 is able to access the disease compartments in various organs and reduce tumour burden.

Combinations are routinely used to treat haematological malignancies to drive durable anti-tumour responses in patients. In order to broaden the activity of AZD0466, we tested clinically relevant standard of care as well as targeted therapies in combination with AZD0466 in DLBCL models with limited Bcl-2 inhibitor single agent activity. In the SUDHL-4 GCB

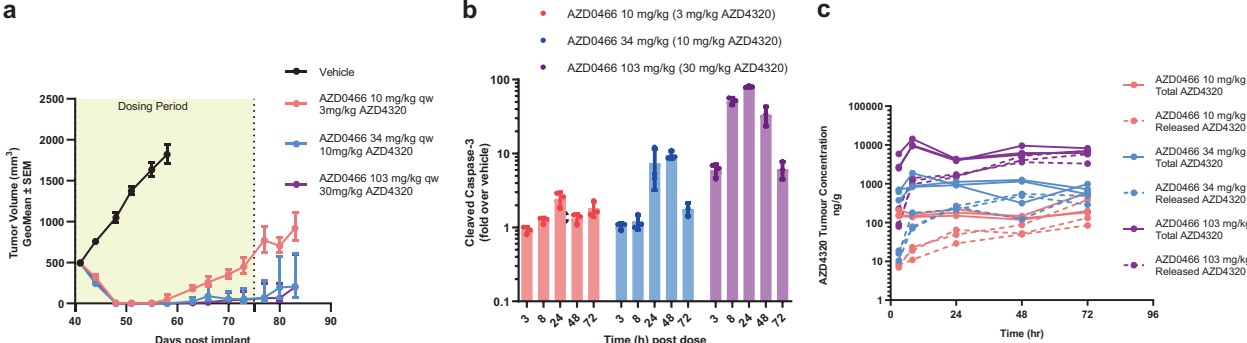

**Fig. 6 Efficacy of AZD0466 in RS4;11 tumour xenograft model. a** Tumour growth inhibition with AZD0466 treatment at 10 mg/kg (red), 34 mg/kg (blue) and 103 mg/kg (purple) doses. Data in the graphs are represented as mean tumour volumes ± SEM (*n* = 6 per group). **b** Kinetics of cleaved caspase 3 induction in tumours after a single administration of 10 mg/kg (red), 34 mg/kg (blue) and 103 mg/kg (purple) AZD0466 doses. Data in the graphs are represented as mean ± SD (*n* = 3 per time point per group). **c** Tumour concentration of total (solid lines) and released (dotted lines) AZD4320 over time following a single dose of 10 mg/kg (red), 34 mg/kg (blue) or 103 mg/kg (purple) of AZD0466. Data in the graphs are represented as mean ± SD (*n* = 3 per time point per group).

DLBCL model, AZD0466 combined at various doses with 10 mg/kg rituximab demonstrated combinatorial benefit and tumour regressions were seen in the combination groups at all doses of AZD0466 studied (Fig. 8c). Kaplan-Meier survival curve analysis showed that 5/6 and 6/6 animals achieved complete and durable regressions at 103 mg/kg and 172 mg/kg doses respectively (Fig. 8d). Tyrosine kinase inhibitors have been shown to "prime" cancer cells for apoptosis by upregulating pro-apoptotic BH3 proteins like Bim[27] making them vulnerable to an inhibitor of Bcl-2 family proteins like AZD0466. To demonstrate this priming effect, the combination of AZD0466 with a BTK (Bruton's Tyrosine kinase) inhibitor, acalabrutinib, was characterized in OCI-LY10 DLBCL model. While neither agent showed any demonstrable monotherapy activity the combination resulted in complete regressions in 8/8 mice in this model (Fig. 8e) and prolonged the survival of these animals (Fig. 8f). These results highlight the ability of AZD0466 combinations to increase the depth and duration of response.

## Discussion

Dual Bcl-2/Bcl-x$_L$ inhibitors are expected to deliver therapeutic benefit in many haematological cancers and solid tumours but have been thus far limited by tolerability issues. The present study sought to design a drug-dendrimer conjugate to improve the therapeutic index of the dual Bcl-2/Bcl-x$_L$ inhibitor, AZD4320. Mathematical modelling was applied to identify an optimal release rate of AZD4320 from the dendrimer carrier in order to maximise the therapeutic effect of AZD4320 in tumours while minimising plasma concentration and cardiovascular toxicity.

To our knowledge, this is the first application of mathematical modelling, to obtain the desired release rate, a critical parameter in nanomedicine design which is imperative to ensure optimal therapeutic index. Given the complexity of whole body PB-PK models for therapeutic nanomedicines where the drug needs to be released for activity, the challenges of accurate bioanalysis of free drug from nanomedicines[28], and the extreme physicochemical properties coupled with unusual pharmacokinetic properties of the drug, a simple semi-empirical mathematical model was developed. This was used to efficiently guide the design of our drug-dendrimer conjugate for optimal release rate.

An initial round of synthesis yielded three drug-dendrimer conjugates with a wide range of release half-lives and demonstrated the importance of release half-life on tumour

concentration, and efficacy, with the faster releasing linkers showing greater activity. Conversely, cardiovascular tolerability, a key dose limiting toxicity, was improved with the slower releasing drug-dendrimer conjugates. This dependence on release rate for optimal efficacy and safety is consistent with other types of drug-dendrimer conjugates[19]. The pharmacokinetic data from these initial conjugates was used to parameterise the model and derive a set of equations to describe the concentrations of conjugated and released drug in both plasma and tumour tissue.

The assumptions of the model are that the drug-dendrimer conjugate is predominantly confined to the vasculature, which is supported by the volume of distribution (59 ml/kg), and is cleared primarily through organs of the RES[29], that in the tissues of interest (i.e. plasma and tumour), AZD4320-dendrimer conjugate is released by hydrolysis at the same rate as in plasma in vitro (Supplementary Fig. 8) and that AZD4320 is delivered to the tumour only by extravasation of the drug-loaded dendrimer into the tumour interstitial fluid and subsequent linker hydrolysis and remains there. This final assumption around AZD4320 delivery to tumour only being via the dendrimer conjugate is unusual. It is supported by preclinical studies which show that AZD4320 delivery to tumours is highly dependent on formulation. Formulations containing hydroxypropyl-β-cyclodextrin led to a much greater tumour concentration and efficacy that alternative formulations despite showing similar plasma concentrations. (Supplementary Fig. 2). The AZD4320 drug itself, due to its very low solubility and high protein binding, does not distribute to the tumour without an enabled formulation. The presence of cyclodextrin-drug complex is thought to lead to improved kinetics of exchange between plasma and the tumour. The combination of a high degree of cyclodextrin and protein binding and the short plasma half-life are thought to be responsible for this effect. Conjugating AZD4320 to a dendrimer both provides enhanced solubility and allows the dendrimer to act as a carrier for the delivery of AZD4320 to tumours. Therefore, delivery of AZD4320 to the tumour when released from the dendrimer carrier in the plasma is unlikely to happen and released drug access to the tumour is assumed to be negligible. This would not be expected for other drugs with more typical physicochemical properties or slower plasma clearance. In these cases, the model would also need to account for delivery of released drug in the plasma to the tumour as well as that delivered via the drug-dendrimer conjugate. This final assumption, also assumes that once the drug-dendrimer conjugate has reached the tumour it is largely retained there. Released concentrations of AZD4320 from

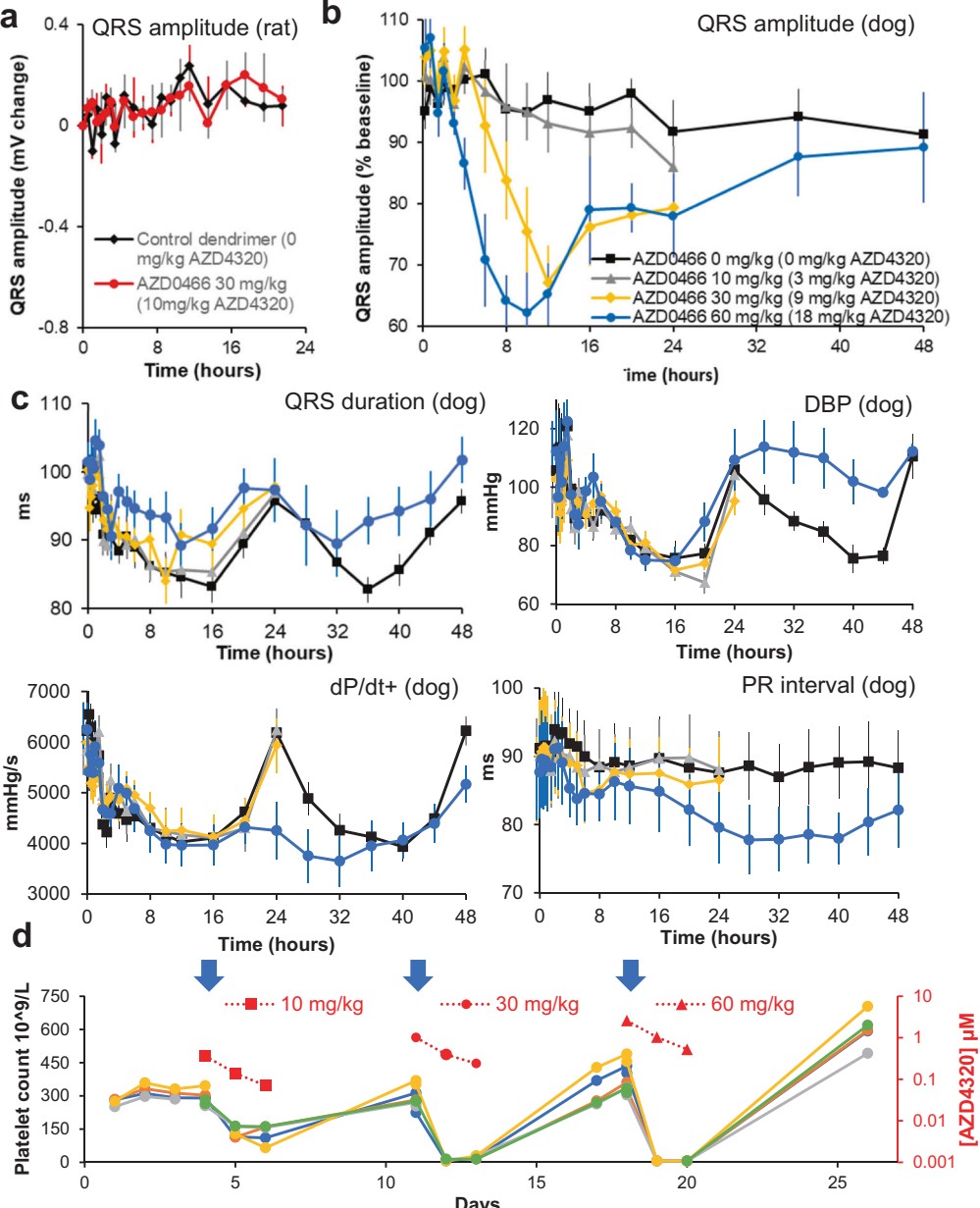

**Fig. 7 Effects of AZD0466 on cardiovascular parameters and platelets in rats and dogs.** Time course of cardiovascular effects in telemetered animals following single dose IV administration at $t = 0$ h of vehicle (0 mg/kg) or ascending dose levels of AZD0466. **a** Effects on QRS amplitude in rat. 0 mg/kg (black diamonds), AZD4320 (red circles). **b** Dose-response effects on QRS amplitude in dog. (c) Dose-response to other cardiovascular parameters that showed a response to AZD0466 (QRS duration, diastolic blood pressure (DBP), change in left ventricular pressure with time (dP/dt+) and PR interval). For **b** and **c**, 0 mg/kg (black squares), 10 mg/kg AZD0466 (grey triangles), 30 mg/kg AZD0466(orange diamonds), 60 mg/kg AZD0466 (blue circles). **d** Time course of changes in platelet count for each of the 4 dogs following single dose IV administration of ascending doses of AZD0466 (left vertical axis). The blue arrows indicate the day of dosing. The time course of AZD4320 concentration in the plasma (defined as the sum of protein bound and unbound AZD4320 which is not dendrimer conjugated) is also shown (right vertical axis, red symbols). Cardiovascular data are group mean ± SEM, $n = 3$ rats per group or $n = 5$ dogs per group.

SPL-8931 continue to increase for up to 48 h, consistent with there being a reservoir of drug-dendrimer effectively trapped and slowly releasing AZD4320 at the tumour site. Retention in the tumour has also been observed via mass spectrometry imaging for similar dendritic delivery systems[19] and other nanoparticles[16].

Experimental in vitro release data, plasma pharmacokinetics and tumour pharmacokinetic data generated from the initial drug-dendrimer conjugates were used to parameterise the model in terms of reticuloendothelial uptake rate ($k_{res}$), tumour extravasation rate ($k_{ext}$) and return of AZD4320 from tumour to circulation ($k_{tu}$). All the drug-dendrimer conjugates had similar molecular weight, size and surface properties so were assumed to behave in a similar manner.

Interestingly, $C_{max}$ in the tumour tissue was delayed as the release half-life increased (Fig. 5c and g) and is predicted to be largely independent of the release half-life from about 5 to 24 h (Fig. 5d). This would not be expected if the drug-dendrimer conjugate was providing a controlled release of the drug to the plasma and the main drug delivery route to the tumour was via this released drug in plasma rather than being through

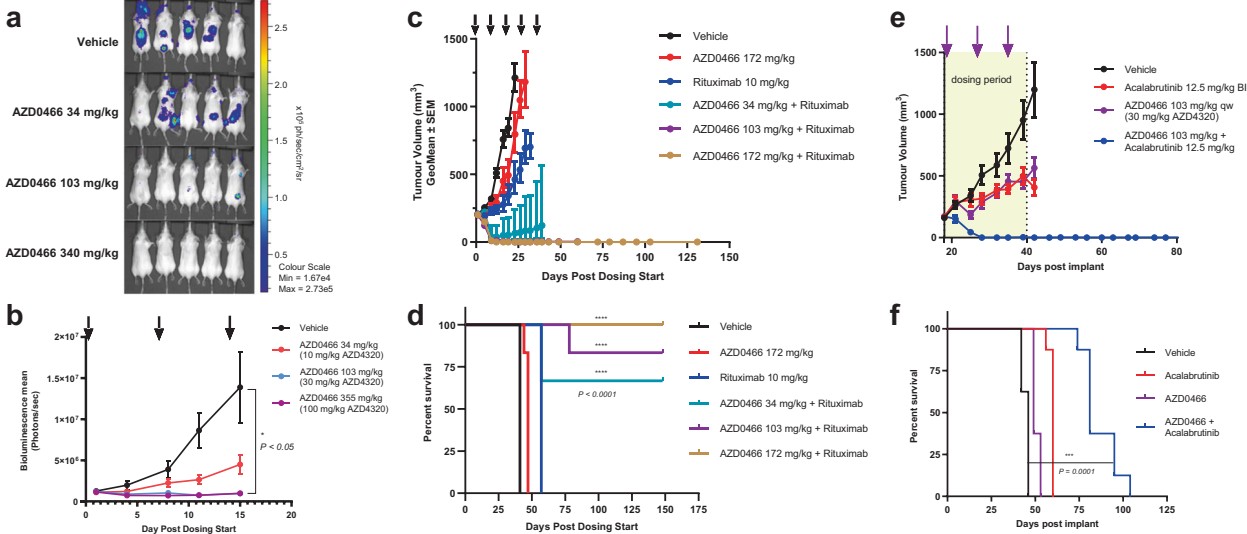

**Fig. 8 AZD0466 anti-tumour activity in Non-Hodgkins lymphoma models. a** In vivo bioluminescence images in disseminated luciferase-tagged Ri-1 bearing mice treated with vehicle and various doses of AZD0466. **b** Tumour burden quantified as photons/sec over time via IVIS imaging in vehicle and treatment groups. Data in the graphs are represented as mean bioluminescence ± SEM ($n = 5$ per group). **c** Tumor growth inhibition in SUDHL-4 xenograft model treated with vehicle (black), rituximab (10 mg/kg, blue), AZD0466 (172 mg/kg, red) and various doses of AZD0466 in combination with rituximab on a weekly schedule for 5 weeks. Data in the graphs are represented as mean tumour volumes ± SEM ($n = 6$ per group). **d** Kaplan-Meier survival curve of vehicle and treatment arms in SUDHL-4 xenograft model. **e** Tumor growth inhibition in OCI-LY10 xenograft model treated with vehicle (black), acalabrutinib (12.5 mg/kg BID, red), AZD0466 (103 mg/kg QW, purple) and AZD0466 in combination with acalabrutinib (blue) for 3 weeks. Data in the graphs are represented as mean tumour volumes ± SEM ($n = 8$ per group). **f** Kaplan-Meier survival curve of vehicle and treatment arms in OCI-LY10 xenograft model.

extravasation of the drug-dendrimer conjugate. If the concentrations in the tumour tissue were driven by the released concentration in the plasma, the model would predict the relative concentrations of released AZD4320 as shown in Supplementary Fig. 15 where the faster releasing dendrimers would lead to a higher tumour $C_{max}$ with all the peak concentrations occurring at the same time. In addition, the tumour $C_{max}$ as a result of dosing the faster releasing SPL-8933 would be predicted to be almost double that of SPL-8932 rather than lower than that of SPL-8932 as observed and predicted by the model.

Concordance between predicted and measured tumour concentration profiles for all six conjugates (initial three and subsequent round of synthesis, Fig. 5c and g, Supplementary Figs. 10 and 11) also provide strong support for the model assumptions. This suggests that extravasation of the drug-dendrimer conjugates from the bloodstream is important for accumulation and retention of AZD4320 within the tumours. It should be noted that $k_{ext}$, the extravasation constant, is only present in the model as a scaling factor and thus this model should be applicable across other tumour types.

The model further identified an optimal release half-life range (20 to 30 h) for simultaneous achievement of adequate released concentration of AZD4320 in the tumour for efficacy, whilst minimising plasma released $C_{max}$ to reduce impact on QRS amplitude and other toxicities driven by plasma $C_{max.}$ and minimising dose (Fig. 5f). Shorter release half-lives will lead to high plasma levels and expected lower tolerability whereas very long half-lives will require higher doses of the drug-dendrimer conjugate or more frequent dosing which would also result in less efficient delivery (fraction of dose reaching the tumour). Efficiency of delivery is important from cost of goods, minimising polymer dose to the patient and patient acceptability perspective in terms of number of vials and frequency of dosing. The second round of synthesis was carried out to target a hydrolysis release half-life in this range; one of these subsequent drug-dendrimer

conjugates yielded a release half-life of 25.5 h (SPL-8977 subsequently named AZD0466) within this optimal range.

AZD0466 exhibits potent and dose dependent anti-tumour efficacy as a single agent in the Bcl-2 dependent RS4;11 subcutaneous model (Fig. 6a). The antitumor activity was accompanied by activation of the mitochondrial apoptosis pathway as measured by induction cleaved caspase. The tumour exposure of released AZD4320 showed good dose linearity. Although all three doses tested resulted in regression in this sensitive model, the durability of response was prolonged at higher doses of AZD0466 indicating the potential for driving deeper responses in the clinic while maintaining tolerability.

As predicted, efficacy of AZD0466 was also similar to the faster releasing construct SPL-8932 at an equivalent dose (Supplementary Fig. 14). From a safety perspective, AZD0466 gave no reduction in QRS amplitude at 30 mg/kg in rat (Fig. 7a) as compared to the faster releasing dendrimer conjugate SPL-8932 where there was a marked decrease in QRS amplitude (Fig. 4b) and to AZD4320 itself (Fig. 4a). Thus, assisted by mathematical modelling, we were able to optimise release half-life and linker design to improve cardiovascular tolerability of our AZD4320-dendrimer conjugate with no loss in efficacy.

Data indicate that, for equivalent efficacy, approximately twice the dose of AZD4320 is required to achieve similar tumour regression when AZD4320 is administered as the dendrimer conjugate, AZD0466. However, tumour regrowth is slower with the dendrimer conjugate (Supplementary Fig. 14). Tumour regression is believed to be driven by equivalent maximum concentrations of free AZD4320 in the tumour. According to model predictions, these similar maximum tumour concentrations result in ~25-fold lower maximum plasma concentrations of released AZD4320 when delivered via the dendrimer conjugate, AZD0466. Indeed, at equivalent AZD4320 doses in the rat (10 mg/kg), AZD4320 gave a 40% decrease in QRS amplitude in comparison to negligible effect on QRS amplitude for AZD0466 (Fig. 7a). Furthermore, in dog,

10 mg/kg of AZD0466 (equivalent to 3 mg/kg AZD4320) had no effect on QRS amplitude and was tolerated up to 60 mg/kg (equivalent to 18 mg/kg AZD4320) (Fig. 7b), whereas 1 mg/kg AZD4320 decreased QRS amplitude by ~70% and was not tolerated in the dog (Supplementary Fig. 1), giving an 18-fold improvement in cardiovascular tolerability based on dose. AZD0466 shows dose-dependent transient thrombocytopenia, and on target toxicity for Bcl-x$_L$ however this can be mitigated via a weekly dosing schedule which allows for platelet recovery between doses (Fig. 7d). The no observed effect level (NOEL) for AZD0466 in dog was determined as 10 mg/kg (equivalent to ~3 mg/kg AZD4320) much higher than 0.2 mg/kg for AZD4320 itself (Supplementary Table 4). These safety studies clearly demonstrate an improved safety profile in both species.

Utility of mathematical modelling to design a nano delivery system is an effective tool for reduced consumption of resources in iterative rounds of synthesis and in vivo profiling, and is also beneficial from the perspective of reducing, refining and replacing animal studies (3Rs)[30]. The generation of robust in vivo pharmacokinetic and biodistribution data is acknowledged as challenging in cancer nanomedicine[31], primarily because of the terminal nature of sampling (necessitating pooled data) and bioanalytical quantification challenges relating to post sample stabilisation and differentiation between released (protein bound and unbound) and conjugated/encapsulated drug. Measurement of concentration kinetics at the target site and in the main toxicological organs is challenging yet imperative in the design and development of any nanomedicine and mathematical modelling plays an important role in driving this understanding and design. Integration and development of mathematical modelling techniques with experimental investigation of nanoparticle kinetics, efficacy and toxicity in order to better understand structure-activity relationships of nanocarriers is important for successful clinical translation of nanomedicines.

The mathematical model developed in this work can be extended to other drug-polymer conjugates and to nanoparticles where the drug is encapsulated, to help inform their design and select an appropriate release rate. In addition, it could be adapted and used to understand and probe factors affecting biodistribution such as size and PEG density affecting the in vivo performance of nanoparticles. A similar physiologically based-pharmacokinetic (PB-PK) approach has been to used describe the drug delivery properties important for nanoparticles fabricated from the polylactide-poly (ethylene glycol) block-copolymer where drugs are encapsulated within the nanoparticle[32]. A tumor-compartment bearing PB-PK model has recently been developed to investigate the effects of nanoparticle properties, tumour variables and individual physiological differences on the plasma circulation time, MPS sequestration, tumor delivery, and excretion of nanoparticles. This model provides important mechanistic information and evaluates the impact of both physiological and pathophysiological conditions that could affect tumour delivery and inform patient selection, however it does not address drug release[33]. Using non-invasive in vivo imaging enables the visualization and quantification of the whole-body disposition behavior of nanoparticles and other advanced therapeutics which is essential for a comprehensive understanding of their distribution. The use of image-guided mathematical modelling for pharmacological evaluation of nanomaterials has been recently reviewed by Dogra et al.[34]. An integrated SPECT/CT imaging-based pharmacokinetics mathematical model has also been explored to evaluate the disposition of mesoporous silica nanoparticles[35]. Mass spectrometry imaging (MSI) is a powerful label-free tool which can map the nanocarrier, released drug and pharmacodynamic effect in tissues and thus can provide critical additional data to deepen mechanistic

understanding[36,37]. Quantitative non-invasive in vivo imaging can help guide the development and parameterization of mathematical models for both descriptive and predictive purposes in nanomedicine design and development.

Additional studies were carried out with AZD0466 to further probe its efficacy and tolerability and suitability for clinical development. AZD0466 also demonstrated efficacy in a disseminated DLBCL model decreasing tumour burden in multiple locations. Furthermore, enhanced efficacy was observed when AZD0466 was combined with standard of care chemotherapy, rituximab, and acalabrutinib, demonstrating the potential of AZD0466 to safely and effectively combine with other therapies to drive deep and durable therapeutic response.

The AZD4320-dendrimer conjugate, AZD0466, identified in this study through mathematical modelling and design-led thinking has the potential to increase therapeutic index of a promising dual Bcl-2/x$_L$ inhibitor for the treatment of haematological and solid tumours. Delivering via a drug-dendrimer conjugate has enabled progression of an important dual Bcl-2/x$_L$ inhibitor into clinical development. A GLP toxicity package has been completed and an investigational new drug application has been approved by FDA and phase I clinical studies are in progress (NCT04214093).

## Methods

**Synthesis and characterisation of AZD4320-dendrimer conjugates**. Methods of preparation of AZD4320 can be found in U.S. Patent No. 9,018,381. Methods of preparation of the fifth-generation (G5) PEGylated dendrimers can be found in WO 2012/167309 A1. The structure is shown in Supplementary Fig. 3. AZD4320 was conjugated through the primary alcohol to the PEGylated dendrimer via different aliphatic linkers to yield drug-dendrimer conjugates (Supplementary Methods).

AZD4320-dendrimer conjugates were characterised with respect to molecular weight, particle size, AZD4320 load, and in vitro release kinetics (Supplementary Methods). In vitro release half-life was calculated by least squares fitting to a 1st order release model.

**Study design**. All animal studies were conducted in accordance with the AstraZeneca Global Bioethics policy. All experimental work is outlined in project license 40/3483 which has gone through the AstraZeneca Ethical Review Process. Studies in the United States were conducted in accordance with the guidelines established by the internal IACUC (Institutional Animal Care and Use Committee) and reported following the ARRIVE (Animal Research; Reporting In Vivo experiments) guidelines.

**Pharmacokinetics studies**. Pharmacokinetic studies were carried out in the female C.B-17 SCID mouse. Either freshly prepared solutions of AZD4320 in 30% hydroxypropyl-β-cyclodextrin or drug-dendrimer conjugates in phosphate buffered saline were injected as a bolus into the tail vein at dosing volume of 5 ml/kg. Blood samples were collected at different time points post dose and analysed as below.

**Bioanalysis of AZD4320 in plasma and tumour samples**. Blood samples were collected from the jugular vein by venepuncture into K$_2$EDTA (as anticoagulant) tubes. Immediately following collection into the K$_2$EDTA tube the samples were mixed gently and kept on crushed ice until centrifugation. Within 30 mins of blood collection, each sample was centrifuged (1500 g for 10 min at 4 °C) to generate plasma which was kept on wet ice at all times in polypropylene tubes. Duplicate 50 μL aliquots were transferred immediately after centrifugation into uniquely labelled polypropylene tubes ('released' AZD4320) containing equivalent amount (50 μL) of 0.2 M citric acid solution, inverted 10 times and then stored in a freezer set to maintain a temperature of −80 °C, until analysis.

The concentrations of 'total' AZD4320 were determined in plasma by overnight incubation with hydrazine followed by protein precipitation with acetonitrile and subsequent liquid/liquid extraction. Liquid chromatography with mass spectrometric detection (LC-MS/MS, details in Supplementary Table 5) was used to quantitate both 'released' and 'total' AZD4320.

For tumour measurements, tumours were harvested, split into smaller pieces and snap frozen in liquid nitrogen. Frozen tumour samples were subsequently weighed and then homogenised for both 'released' and 'total' AZD4320. For 'released' AZD4320, samples were homogenised on ice with an equal volume (w/v) of blank plasma containing 0.2 M citric acid. The samples were then processed according to the aforementioned plasma approach. For 'total' AZD4320, samples

were homogenised with an equal volume of plasma (w/v) and then processed according to the aforementioned 'total' approach.

Sample analyses were performed in batches containing study samples, calibration standards at 10 concentration levels, QC samples at three concentration levels in duplicate and blank samples. The analytical run acceptance criteria for calibration standards and QC samples was that at least 75% of back-calculated concentrations for calibration standards and at least 2/3 of the QC samples (at least 50% at each concentration level) should be within ±20% (±25% at LLOQ) from nominal value.

**Mathematical modelling**. The model is based on four key assumptions: (i) The dendrimer is mostly confined to the vasculature and is cleared primarily through RES[29], (ii) In the tissues of interest (plasma and tumour), dendrimer-conjugated AZD4320 is released by hydrolysis at the same rate as that measured in vitro at 37 °C and pH 7.4. (iii) AZD4320 is delivered to the tumour by extravasation of the drug-loaded dendrimer into the tumour interstitial fluid and subsequent linker hydrolysis (iv) Dendrimers reaching the tumour interstitium through extravasation are effectively retained (as a result of the EPR effect[38]) such that there is negligible backflow of dendrimer bound drug into the blood.

In support of assumption (iii), delivery of released AZD4320 from the plasma to the tumour without the dendrimer as a carrier is considered negligible based on its dependence on formulation for efficacy (Supplementary Fig. 2). Evidence for assumption (iv) is demonstrated by the released concentrations of AZD4320 from SPL-8931 which continue to increase up to 48 h, consistent with there being a reservoir of dendrimer conjugated AZD4320 effectively trapped for a long period of time at the tumour site. Retention in the tumour has been observed for a similar dendritic delivery systems[19].

For the case of a bolus dose of dendrimer-conjugate, closed form mathematical expressions for the concentration of (i) released AZD4320 in tumour tissue, (ii) dendrimer-conjugated AZD4320 tumour tissue and (iii) the concentration of dendrimer bound AZD4320 in plasma have been derived (Supplementary Methods) and are summarised below (Eqs. (1)–(4)).

The concentration of dendrimer conjugated AZD4320 in plasma as a function of time $C_{c,pl}(t)$ (compartment 1 in Fig. 5a) is given by,

$$C_{c,pl}(t) = \frac{D}{V_{pl}} \exp(-(k_{res}+k_h)t) \qquad (1)$$

where $D$ denotes the dose of dendrimer-bound drug, assuming that all drug is dendrimer bound initially, (as AZD4320 equivalent), $k_{res}$ is a constant which accounts for RES uptake of the dendrimer, $V_{pl}$ is the volume of plasma and $k_h$ is the first order linker hydrolysis rate constant under biorelevant conditions of 37 °C at pH 7.4.

The concentration of dendrimer conjugated AZD4320 in the tumour tissue (compartment 2 in Fig. 5a) as a function of time, $C_{c,tu}(t)$, can be expressed as,

$$C_{c,tu}(t) = \frac{D}{V_{tu}} \frac{k_{ext}}{k_{res}} \exp(-k_h t)(1-\exp(-k_{res}t)) \qquad (2)$$

where $k_{ext}$ is a constant which determines the degree of extravasation of dendrimers from the vasculature into the tumour interstitium and $V_{tu}$ is the volume of the tumour.

Finally, the concentration of released active (compartment 3 in Fig. 5a) at the tumour site as a function of time, $C_{r,tu}(t)$, can be expressed as,

$$C_{r,tu}(t) = \frac{Dk_{ext}k_h}{V_{tu}k_{res}}\left(\frac{k_{res}\exp(-k_{tu}t)}{(k_h-k_{tu}+k_{res})(k_h-k_{tu})} + \frac{\exp(-(k_h+k_{res})t)}{(k_h-k_{tu}+k_{res})} - \frac{\exp(-k_h t)}{(k_h-k_{tu})}\right) \qquad (3)$$

where $k_{tu}$ is a rate constant relating to the return of released drug to the systemic circulation (which was determined by fitting to experimental data from the initial set of dendrimer conjugates, i.e., SPL-8931, SPL-8932 and SPL-8933). Equation (3) was used to assess the likely impact of the hydrolysis rate on efficacy.

Parameters were derived by fitting the appropriate equations to the initial dataset using custom Python code (Python 3.7.4, SciPy 1.3.1)[39].

The released AZD4320 concentrations based on the model, as a function of time can be very well approximated (at least for times in the domain $0 < t \le 2$ h) by the following expression,

$$C_{r,pl} \approx \frac{D}{V_c}\left(\frac{k_h(\exp(-((CL/V_c)+k_{12}+k_{13})t)-\exp(-(k_h+k_{res})t))}{k_h+k_{res}-(CL/V_c+k_{12}+k_{13})}\right) \qquad (4)$$

where $CL$ and $V_c$ are the total plasma clearance and central volume of distribution for the released AZD4320 (determined from dosing a conventional IV formulation of AZD4320) respectively and $k_{12}$ and $k_{13}$ are compartmental rate constants (also from the conventional IV dose of AZD4320).

Equation (4) was used to assess the impact of the hydrolysis rate on the plasma $C_{max}$ of the released active moiety, AZD4320 and the resulting CV tolerability.

To obtain a term for the optimisation index (Fig. 5f), it is assumed that the therapeutic index (TI) is dependent on the ratio of the tumour $C_{max}$ per unit dose

to the plasma $C_{max}$ per unit dose,

$$TI\left(t_{1/2}\right) \propto \frac{\bar{C}_{tu,max}\left(t_{1/2}\right)}{\bar{C}_{pl,max}\left(t_{1/2}\right)} \qquad (i)$$

where, $\bar{C}_{tu,max}\left(t_{1/2}\right)$ denotes the predicted tumour maximum concentration (as a function of release half-life) of released AZD4320 per unit dose and $\bar{C}_{pl,max}\left(t_{1/2}\right)$ is the predicted plasma maximum concentration of released AZD4320 per unit dose (as a function of release half-life).

To account for the impact of release half-life on total dose, the effective dose must be adjusted inversely proportionally to the predicted tumour $C_{max}$ per unit dose, if equivalent values of tumour $C_{max}$ are to be achieved. Thus, for a constant tumour $C_{max}$,

$$D_{eff}\left(t_{1/2}\right) \propto \frac{1}{\bar{C}_{tu,max}\left(t_{1/2}\right)} \qquad (ii)$$

Combining (i) and (ii) leads to, an overall optimisation index (λ) which takes therapeutic index and required dose into account

$$\lambda\left(t_{1/2}\right) = \frac{\bar{C}^2_{tu,max}\left(t_{1/2}\right)}{\bar{C}_{pl,max}\left(t_{1/2}\right)} \qquad (iii)$$

Release half-lives which give the highest values for λ, represent the optimal choice of dendrimer in terms of therapeutic index and overall dose requirements.

**Xenograft studies**. Female C.B.−17 SCID mice were purchased from Charles River Laboratories. Mice were 5–8 weeks old at the time of tumour implantation. All xenograft studies were conducted at our Association for the Assessment and Accreditation of Laboratory Animal Care accredited facility in Waltham, MA in accordance with ethical regulations described in the guidelines established by the internal Institutional Animal Care and Use Committee. The dendrimer conjugates were formulated in citrate/phosphate buffer pH 5.0, diluted 1:10 with 5% glucose containing 1% w/v Kolliphor HS15, and AZD4320 was formulated in 30% HP-β-CD and dosed as a weekly intravenous (IV) administration at a volume of 5 ml/kg at the indicated doses. Five million RS4;11, five million SUDHL-4 or five million OCI-LY-10 cells were injected subcutaneously in the right flank of mice in a volume of 0.1 ml. Tumour volumes (measured by caliper), animal body weight, and tumor condition were recorded twice weekly for the duration of the study. The tumour volume was calculated (taking length to be the longest diameter across the tumour and width to be the corresponding perpendicular diameter) using the formula: length (mm) x width (mm)²/0.52. For efficacy studies, mice were randomized based on tumour volumes using stratified sampling and enrolled into control and treatment groups. Tumour growth inhibition from the start of treatment was quantified by measuring the difference in tumour volume between control and treated groups. Survival analysis was evaluated using Mantel-Cox test. Tumours were also taken for bioanalysis.

**Disseminated efficacy study**. Ten million luciferase-tagged Ri-1 tumour cells were injected intravenously in the tail vein of C.B.−17 SCID female mice in a volume of 0.1 ml. Tumour burden (determined by Xenogen IVIS imaging), body weight and health condition were recorded twice weekly for the duration of the study. For Xenogen imaging, mice received an intraperitoneal administration of 150 mg/kg D- Luciferin 10 to 15 minutes prior to imaging. Imaging was performed under isoflurane anaesthesia and data analyzed using Living Image software (Xenogen). Statistical significance was evaluated using non-parametric, unpaired, two-tailed $t$-test.

**Measurement of cleaved caspase 3**. Tumours were excised, snap frozen in liquid nitrogen and stored at –80 °C. Tumours were homogenised in lysis buffer containing protease and phosphatase inhibitors, protein concentration determined by Pierce BCA Protein Assay Kit and diluted in lysis buffer to a protein concentration of 0.4 μg/μL. Cleaved caspase-3 (CC3) was measured using the mesoscale cleaved caspase-3 (Asp175) assay whole cell lysate kit according to manufacturer's instructions. Induction of CC3 was expressed as fold change of treated tumours over vehicle control.

**Immunohistochemistry**. Formalin fixed paraffin embedded (FFPE) slides were cut on a rotary microtome at 4 μm and placed onto Superfrost plus slides (VWR) dried thoroughly, and then baked for 20 min in an oven at 60 °C. Slides were then stained on a Ventana Discovery XT using a CCI standard protocol and DAB Map Kit (Ventana cat # 760-124). The slides were then incubated in the primary CC3 (Cell Signaling Technology) antibody at 100μL per slide of a 1:50 dilution for 1 h. The secondary antibody was a biotinylated goat anti-rabbit from Vector Labs (PK-6101) made according to manufacturer's instructions and put in an auto-dispense container. The slides were incubated in the secondary antibody for 30 min and then counterstained with Hematoxylin (Ventana 760-2021) for 4 min and blued with bluing reagent (Ventana 760-2037) for 4 min. After the Ventana run was complete, slides were taken off the machine, rinsed in reaction buffer, dehydrated,

cleared and mounted with permount. Digital slide images were acquired at 20 x magnification using the Aperio Scanscope XT (Vista, CA).

**Conscious dog telemetry**. Experiments to assess the cardiovascular effects of AZD4320 and AZD0466 were performed using methodology previously described[40]. Animals were instrumented with telemetry devices (Data Sciences International, St. Paul, MN USA) for use in the study. Whilst under anaesthesia, the telemetry transmitter was placed in the abdominal muscle and the arterial blood pressure catheter was placed in the femoral artery and advanced to the abdominal aorta. The left ventricular pressure catheter was inserted through the apex of the heart into the left ventricle. The ECG electrodes were sutured onto the heart, one at the apex and one at the base.

Arterial blood pressure, heart rate, left ventricular pressure, lead II ECG (including QRS amplitude - the combined maximum positive and negative deflections of the QRS complex) were recorded in conscious telemetered male beagle dogs for at least 1 h pre-dose and up to 48 h post-dose. Due to variation in QRS amplitude between dogs, this parameter is presented at % baseline. Recordings took place in a telemetry pen via receivers placed inside the dog pens, with animals free to move except during dosing and blood sampling procedures. Telemetry signals were acquired and analysed using Ponemah software, version 5.0 and 5.1 (St. Paul, Minnesota, USA).

All experiments used an escalating dose design, with an interval of 7 days between treatments. For evaluation of AZD0466, $n = 5$ dogs received single doses of control dendrimer (0 mg/kg AZD0466) or 10, 30 and 60 mg/kg AZD0466, administered as a 1-hour intravenous infusion. For AZD4320, $n = 2–4$ dogs received single intravenous doses of 0 (vehicle: 20% Captisol, pH 9-9.06) and 0.2, 0.85, 1 or 1.7 mg/kg AZD4320 infused over 0.5 h or 0.85 mg/kg AZD4320 infused over 3 h.

**Conscious rat telemetry**. Experiments to assess the cardiovascular effects of AZD4320, AZD0466 and other drug-dendrimer conjugates were performed using methodology previously described[41]. Male Han Wistar rats were implanted with HD-S11 transmitters (Data Sciences International, St. Paul, MN USA) under isoflurane anaesthesia, the telemetry transmitter was placed in the peritoneal cavity, the arterial blood pressure catheter was placed in the terminal abdominal aorta and the ECG electrodes were sutured, one at the dorsal surface of the xiphoid process and one at the anterior mediastinum, close to the right atrium[42]. Animals were allowed to recover for at least 2 weeks and were around 2–3 months at the time of experimentation. The animals were pair-housed in standard rat cages with a non-instrumented companion throughout the study. Cages were prepared with bedding material and enrichment.

Telemetry recordings took place in the home cage. Arterial blood pressure, heart rate, and ECG (including QRS amplitude - the combined maximum positive and negative deflections of the QRS complex), were recorded continuously via receivers placed beneath the home cage for at least 1-hour pre-dose and for up to 24 h post-dose. The animals were removed from the cage temporarily for dosing and blood sampling. Animals were not subject to restraint except where dosing and blood sampling was performed. Data acquisition used Dataquest Open A.R.T.™ with HEM software version 4.2 (Notocord Inc).

For evaluation of AZD4320, $n = 8$ rats received single doses of 0 (vehicle: 30% Kleptose, pH9) or 10 mg/kg AZD4320, administered as a 30-min intravenous infusion. A cross-over study design was used, with an interval of 7 days between treatments. For experiments with AZD0466 and other drug-dendrimer conjugates, separate groups of a minimum of $n = 3$ rats were evaluated. The groups were: control dendrimer, SPL8931 30 mg/kg, SPL8932 30 mg/kg or AZD0466 30 mg/kg, each administered as a 2-minute intravenous infusion.

**Telemetry data analysis**. Data for all parameters were collected continuously during the recording period. The continuous data from each animal was binned into chosen time intervals of varying length and averaged to generate a single data point for each reported nominal time. Each reported nominal timepoint was then averaged across each of the animals tested for each dose level. Based on extensive experience of conducting rat and dog telemetry studies at AstraZeneca, decreases in QRS amplitude is a phenomenon not observed prior to testing AZD4320. Based on analysis of viability of vehicle-treated data, a decrease in QRS amplitude of 15% or more was defined as a compound-related effect.

**Statistics and reproducibility**. Data from the in vivo efficacy experiments are reported as mean ± SD/SEM and the number of animals dosed per group ($n = 3$, $n = 6$ and $n = 8$) are clearly shown in each Figure caption. GraphPad PRISM software (version 8.0) was used for statistical analysis of the efficacy data. Cardiovascular telemetry data focused on the reduction in QRS amplitude. Based on our past experience of evaluation of hundreds of compounds in cardiovascular telemetry studies, a reduction in QRS amplitude is an extremely rare event. However, treatment with AZD4320 caused a reproducible reduction in QRS amplitude in both rats and dogs. Therefore it was possible to screen AZD4320-containing dendrimers for the effect on QRS amplitude using limited animal numbers (typically $n = 3$ rats), as any reduction in QRS amplitude >15% was deemed to be treatment related and unlikely to be a chance event. For dog telemetry studies, a power analysis assessing all cardiovascular endpoints indicated that $n = 4$ dogs is sufficient to show clear compound-related effects, and this number of dogs is used

routinely when assessing cardiovascular safety prior to first-in-human studies. Statistical significance was evaluated using non-parametric, unpaired, two-tailed $t$-test. Survival analysis was evaluated using Mantel-Cox test in Fig. 8.

**Reporting summary**. Further information on experimental design is available in the Nature Research Reporting Summary linked to this paper.

## Data availability
The source data for Figs. 2–8 and Supplementary Figs. S2, S8, S9, S10, S11, S13, S14 & S15 are provided as a Source data file.

## Code availability
Berkeley Madonna, version 8.3.14, University of California at Berkeley was used to generate PK parameters for AZD4320. Custom code was developed, for fitting model equations to experimental data used in the modelling, using Python 3.7.4 and the open source function library Scipy version 1.3.1. Full details of the Python code are available[39]. Dog telemetry signals were acquired and analysed using Ponemah software, version 5.0 and 5.1 (St. Paul, Minnesota, USA). Rat telemetry data acquisition used Dataquest Open A.R.T.™ with Notocord HEM system version 4.2 (Notocord Inc).

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

## Acknowledgements

We thank Marc McCormick, Dave Benstead, Kevin Treacher and Shirley Xia for construct characterisation and method development, and Filippa Shub for review of the manuscript.

## Author contributions

M.B.A. performed conception and design of research, data interpretation and writing and review of the manuscript. C.M.P., K.J.H. performed data interpretation and writing the manuscript. I.G. developed in silico modelling methodology, data interpretation and visualisation and writing the manuscript. S.G., A.R.H., S.R., M.W., F.D.G., and E.G. performed design of research, interpretation of data and writing and review of the manuscript. A.A. and S.W. designed, executed and analysed the in vivo efficacy studies. S.B.B. performed data analysis, interpretation, data visualisation and manuscript writing. J.C. performed in vitro study design, execution and data analysis. C.R. performed design of in vivo studies, oversight and supervision of IHC work. A.G.S. performed design of in vivo studies, oversight and supervision of in vivo work.. W.M., E.H., P.K., J.P., B.K., and M.G. performed conception and design of linkers, as well as prepared drug dendrimer conjugates and developed initial scaled materials and purification methods. M.S. and S.P. performed design of research, animal study design, analysis and interpretation. S.E.F., D.J.O. and P.J.S. performed conceptualisation of research. B.R.D, J.S. and L.G. performed project management activities and review of the manuscript. P.P-D. generated bioanalytical data.

## Competing interests

The authors declare the following competing interests: C.M.P., S.B.B., I.G., P.P-D., W.M., J.P., K.J.H., F.D.G., E.J.H., P.K., A.R.H., S.G., S.P., S.R., M.S., L.G., J.P.S., A.G.S., S.W. A.A., C.R., J.C., M.W., E.G., S.E.F., J.S., B.R.D., M.B.A are current or former AstraZeneca employees and shareholders. D.J.O., M.G., and B.K. are current or former Starpharma employees and shareholders. M.B.A, I.G, E.J.H, W.M, M. G., B.K, D.J.O., J.P.S are inventors of a patent filed related to the therapeutic dendrimers.
