## [Peer Review File · Communications Biology]

Reviewers' comments:

Reviewer #1 (Remarks to the Author):

This is a very interesting paper it clearly shows the importance of the rational design, with an adequate use of the linking chemistry use and drug loading. It was also very interesting to see how mathematical approaches could predict release kinetics, facilitating narrowing down the design alternatives.

It is a very complete manuscript from rational design linker optimization up to in vivo data including biodistribution, pharmacological activity and nanotoxicological profile focusing on cardiotoxicity by means of telemetry

Only minor comments:

1. Could you please discuss the regioselectivity/chemoselectivity of the aromatic NH group through which AZD4320 is derivatised vs the secondary -OH group in its structure?
2. In Fig2b for the release graph, could it be possible to add a longer time point for the 3 conjugates (12 h), just to secure the fittings. And more importantly, could you show the same figure for the newly design linkers SPL-8974, SPL-8976 and SPL-8977 or only SPLC-8977 as it is the relevant compound to fully understand the rational for design. I understand would most probably not be possible to show the linkers of choice, but could you please describe if they are protease, pH or redox dependence in the later conjugates? It will be facilitating the understanding within the readers
3. Could your mathematical modelling be extended to other type of polymer-drug conjugates? How much of a real predictor do you think it should considering? Which is the minimum amount of experimental data you need to feed in the model to be reliable?
4. Have you explored different administration routes in your preclinical models? s.c. vs i.v. or i.p. Could you discuss the expected differences and how is the best way to select the administration route for a polymer-drug conjugate to be studied preclinically.

Reviewer #2 (Remarks to the Author):

In this paper, the authors present a combined experimental and mathematical modeling study focused on designing a dendrimer conjugate of AZD4320 (i.e., AZD0466) for improving (pre)clinical efficacy and cardiovascular tolerability. They have used an iterative approach, including design, synthesis, modeling, and in vivo testing, to optimize the structure of the AZD4320-dendrimer conjugate. As a result, they successfully developed a PK model for predicting the impact of linker hydrolysis rate on the concentration of released AZD4320 in plasma and in tumors, the result of which was then used to guide optimal linker design. Model analysis allowed them to identify an optimal release rate of AZD4320 from the dendrimer carrier in order to maximize the therapeutic effect of the drug in tumors while minimizing cardiovascular toxicity, and they have further tested the new drug-dendrimer conjugate in different animal models. Overall, this is a compelling piece of work. The attempt to integrate mathematical modeling in their drug development pipeline presents an impressive effort. Moreover, the paper is well structured, and it can be seen that the authors tried not to make overstatements throughout the paper. That said, some further analysis can be done, as described below. The reviewer has some other concerns about the paper as well, and would like to ask the authors to consider these before resubmission.

Comments:

1. The authors should do a further literature review, especially on mathematical modeling of cancer in treatment efficacy prediction and optimization. It may be a good idea to discuss different

modeling approaches briefly, which will give the reader a quick understanding of what other modeling approaches are being used in the field and where to place their work.

2. Mathematical model. The assumption for retaining the dendrimers in the tumor interstitium until drug is released should be further justified. What if we allow drug-dendrimers return to the blood stream? Generally speaking, this should be allowed, right?

3. Mathematical model. There is another assumption on the model that needs to be further discussed. That is assumption #3, "delivery of released AZD4320 from the plasma to the tumour without the dendrimer as a carrier is considered negligible". This should be further justified. If not, then does this mean without the dendrimer as a carrier, the drug will not have an impact on the tumor? What about other free drugs in cancer treatment that have been tested in the past, or the free drugs that are currently in use in the clinic?

4. With respect to toxicity, only cardiovascular tolerability was analyzed and discussed. In fact, a much more appropriate approach is to use a whole-body PKPD model to analyze what impact(s) the drug or drug-dendrimer can have on other organs. It may be too late to do this now, and the reviewer understands that this model is a simplified version of a potentially very complex model... At least, the authors may want to address this in Discussion. The authors have cited a couple papers in this field already, e.g., this paper, Dogra et al., PMID: 30949850. Interestingly, this group has recently published a few more; worth having a look, PMID: 32314552, PMID: 32206211, PMID: 30382084, etc.

5. It is interesting to find that the authors have taken dose into account when attempting to define an overall optimization index where the predicted tumor C_{max} to plasma C_{max} ratio is effectively divided by the required dose. Would dosing frequency serve as another factor to be considered in this optimization index as well? At least, the authors may want to provide some discussions about this.

6. Figure 5.

1) Line 292: Fig. 5c should be Fig. 5e.

2) Fig. 5f, where are the other two linker chemistries: SPL-8974 and SPL-8976?

3) While the overall model performance in terms of predicting drug concentration behavior seems acceptable, in (g) it seems that the model only works for SPL-8974, not for SPL-8976 and SPL-8977; in (h), the model only works for the case of Released AZD4320.

7. Figure 6.

Fig. 6b: data for 3mg/kg AZD4320 should be provided as well.

Fig. 6c: data for 3mg/kg AZD4320 should be provided as well.

Communications Biology - Response to Reviewers

Reviewer 1 Comments	Response
1. Could you please discuss the regioselectivity/chemoselectivity of the aromatic NH group through which AZD4320 is derivatised vs the secondary -OH group in its structure?	AZD4320 is derivatized through the primary alcohol, not the aliphatic NH. The regioselectivity is driven by greater reactivity of primary over secondary alcohol. The secondary alcohol was essentially inert under the reaction conditions. The single secondary aromatic amine was also found to be inert under the reactions conditions, thus excellent chemo selectivity was obtained. This amine is expected to be very inert to acylation due to the strongly deactivating influences of both the electron withdrawing ortho-trifluoromethylsulfonyl group and the para-sulfonamide. To make this clearer in the manuscript we have updated Figure 1b to show the linker more clearly and added the following words to the Results section , line 139 “on its primary alcohol” and to the Methods “through the primary alcohol” (line 713).
2. In Fig2b for the release graph, could it be possible to add a longer time point for the 3 conjugates (12 h), just to secure the fittings. And more importantly, could you show the same figure for the newly design linkers SPL-8974, SPL-8976 and SPL-8977 or only SPLC-8977 as it is the relevant compound to fully understand the rational for design. I understand would most probably not be possible to show the linkers of choice, but could you please describe if they are protease, pH or redox dependence in the later conjugates? It will be facilitating the understanding within the readers	Thank you for this comment. We have added a supplementary Figure to show the plasma profiles of all 6 conjugates, Figure S9. An additional sentence has been added in the Results section under Mathematical Modelling sub heading, line 323 to emphasize the similar profiles of the additional drug-dendrimer conjugates. “The plasma profiles of AZD4320 (conjugated and released drug) from these three additional drug-dendrimer conjugates is very similar to those of the initial three drug-dendrimer conjugates, this is shown in Figure S9.” We have now added the sentence “The mechanism of release was hydrolysis”(line 323) to make clear the linkers are design with the same mechanism of release for the second round of synthesis. We have also added a sentence to describe Fig 2 in Results section to emphasise the similar profiles in this initial phase, line 178. “The similar behaviour during the initial phase (up to 6 hours) shows that the release rate only plays a relatively small part in the decline of the total concentrations in plasma and suggests that RES uptake is a predominant factor in clearance and is similar across all these dendrimers that have very different release half-lives.”
3. Could your mathematical modelling be extended to other type of polymer-	The model is not specific to drug-dendrimer conjugates and is also applicable to other polymer drug conjugates and some

	drug conjugates? How much of a real predictor do you think it should considering? Which is the minimum amount of experimental data you need to feed in the model to be reliable?	other nanoparticles (where the drug is encapsulated). The model requires the plasma pharmacokinetics of the free drug, total drug (conjugated and released) plasma pharmacokinetics and tumour pharmacokinetics with the drug polymer conjugates in question. Ideally, a range of release rates are required in order to test the validity of the model assumptions for the selected polymer-drug conjugate.
4.	Have you explored different administration routes in your preclinical models? s.c. vs i.v. or i.p. Could you discuss the expected differences and how is the best way to select the administration route for a polymer-drug conjugate to be studied preclinically.	The focus of our compound design and pre-clinical work has been intravenous delivery as we wanted to use this route of administration in the clinic. Bcl2/Bcl _{x_L} inhibitors are prone to thrombocytopenia and an intravenous dosing schedule where the drug release in plasma is controlled to lower concentrations and the platelets are given time to recover between doses was considered preferable. This is the compound we have progressed to clinical development. The rationale and data to support an intravenous dosing route of administration is discussed in Balachander et al, 2020 “Intermittent targeting of Bcl-2 and Bcl-x_L Induces Tumor Regression that Endures Beyond Transient Thrombocytopenia” which has been just re-submitted to Clinical Cancer Research following revision. We have done some exploratory work using sub cutaneous route of delivery but we feel that this is beyond the scope of this paper. This is not a traditionally used route of administration in oncology. Adverse reactions at the site of administration are a concern with cytotoxic and cell death agents with s.c. administration. IP administration tends to be used for local delivery in ovarian cancers in the clinic but less so for systemic therapy and is not a preferred clinical administration route.
Reviewer 2 Comments		Response
1.	The authors should do a further literature review, especially on mathematical modeling of cancer in treatment efficacy prediction and optimization. It may be a good idea to discuss different modeling approaches briefly, which will give the reader a quick understanding of what other modeling approaches are being used in the field and where to place their work.	Thank you for this helpful suggestion and feedback. We have added to the Discussion with context around the mathematical modelling in the field, the different approaches taken, challenges and the importance of use of mathematical modelling in both the design and development of nanomedicines. This is in two places, initially to provide more context at the beginning (line 496) and later (line 646) in the Discussion. “Physiologically based pharmacokinetic (PB-PK) mathematical modelling is routinely used and critical to understanding the absorption, distribution, metabolism and elimination of drugs and needs to be adapted and used more in nanomedicine design [28]. More diverse mathematical modelling techniques including kinetic and coarse grained molecular dynamic simulations to investigate protein corona formation, continuum and hybrid models to look at microvascular transport, discrete models to evaluate intracellular delivery, pharmacokinetic models for distribution and clearance, hybrid

models for tumour delivery and pharmacodynamic models to describe efficacy and safety have all been instrumental in improving the understanding of nanomaterial behaviour in biological systems for improved therapeutic outcomes in cancer nanomedicine [29] Integration and development of these mathematical modelling techniques with experimental investigation of nanoparticle kinetics, efficacy and toxicity in order to better understand structure-activity relationships of nanocarriers is important for successful clinical translation of nanomedicines. To our knowledge, this is the first application of mathematical modelling, to obtain the desired release rate, a critical parameter in nanomedicine design and imperative to ensure optimal therapeutic index. Given the complexity of whole body PB-PK models for therapeutic nanomedicines where the drug needs to be released for activity, particularly during the design/discovery phase, the challenges of accurate bioanalysis of free drug from nanomedicines, the extreme physicochemical coupled with unusual pharmacokinetic properties of our drug, we elected to focus on developing a simple mathematical model to efficiently guide the design of our drug-dendrimer conjugate. This was focussed on predicting tumour concentrations, as the key effect organ, and minimising plasma concentrations as the surrogate for potential toxicities.”

Line 646:

“Measurement of concentration kinetics at the target site and in the main toxicological organs is challenging yet imperative in the design and development of any nanomedicine and mathematical modelling plays an important role in driving this understanding and design. The mathematical model developed in this work can be extended to other drug-polymer conjugates and nanoparticles where the drug is encapsulated to help inform their design and select an appropriate release rate, as in this study. In addition, it could be adapted and used to understand and probe factors affecting biodistribution such as size, PEG density and release rate affecting the in vivo performance of nanoparticles. A similar physiologically based-pharmacokinetic (PB-PK) approach has been used to describe the drug delivery properties important for nanoparticles fabricated from the polylactide-poly(ethylene glycol) block-copolymer where drugs are encapsulated within the nanoparticle [33]. A generalized and comprehensive tumour-compartment bearing PB-PK model has been recently developed to investigate in silico the effects of nanoparticle properties, tumor variables and individual physiological differences on the plasma circulation time, MPS sequestration, tumour delivery, and excretion of nanoparticles. Analogous to the model used in this work, it has been validated with pre-clinical data and shown to have good predictive capability. This model provides important

		mechanistic information for the design of future nanomedicines and the impact of both physiological and pathophysiological conditions that could affect tumour delivery and inform patient selection [34]. The effects of size, surface chemistry and route of administration on in vivo disposition of mesoporous silica nanoparticles using an integrated SPECT/CT imaging-based pharmacokinetics mathematical model has also been explored [35]. Using non-invasive in vivo imaging modalities enables the visualization and quantification of the whole-body disposition behavior of nanomedicines and other advanced therapeutics which is essential for a comprehensive understanding of their pharmacokinetics and pharmacodynamics. Integrating mathematical modelling and quantitative non-invasive in vivo imaging can also help guide the development and parameterization of mathematical models for both descriptive and predictive purposes and exploiting this can be used to aid the design, development and successful clinical translation of innovative nanomedicines. The use of image-guided mathematical modelling for pharmacological evaluation of nanomaterials has been recently reviewed by Dogra et al. [36]. Matrix-assisted laser desorption/ionization (MALDI) mass spectrometry imaging (MSI) is a powerful label-free tool to map to both the nanocarrier and released drug in tissues and can provide critical additional data that can be used to deepen mechanistic understanding [17] [37].”
2.	Mathematical model. The assumption for retaining the dendrimers in the tumor interstitium until drug is released should be further justified. What if we allow drug-dendrimers return to the blood stream? Generally speaking, this should be allowed, right?	Thank you for this comment. Theoretically, it would be possible for drug loaded dendrimers to return to the blood stream however, however, this is minimal or very slow relative to the timescale of the experiments. One would expect there to be significant rate of input of dendrimer conjugated drug into the tumour from the blood stream for the rather short period of time after dosing < 12 hr where the number of nanoparticles are highest. However, we see high total levels of AZD4320 (dendrimer conjugate and released) persisting in the tumour tissue at much later timepoints (>24 hr) where plasma concentration is low. If the loss of intact dendrimer from the tumour to the bloodstream were significant, we should expect to see, at the later timepoints, concentrations of dendrimer bound drug to decline faster than can be explained by release from the dendrimer. The released AZD4320 concentrations in tumour from dosing SPL-8931 (with a release half-life of 217 hr) continue to increase between 31 hr to 48 hr, consistent with there being a reservoir of dendrimer conjugated AZD4320 effectively trapped at the tumour site. This observation of the drug being retained is supported by previous published work with dendrimers (England et al 2017) and polymeric nanoparticles (Ashton et al 2016) and additional in house data where mass spectrometry imaging

		has shown prolonged retention of drug delivered by dendrimers and nanoparticles in tumours. To qualify the statement in the Mathematical Modelling Materials and Methods section we have deleted indefinitely after “retained” line 775 and added “and observed imaging data” (England et al 2017) to also support the assumption. In addition, an additional sentence has been added to help justify this assumption, line 557 in the Results. “This final assumption also assumes that once the drug has reached the tumour it is retained there, evidence is demonstrated by the released concentrations of AZD4320 from SPL-8931 which continue to increase up to 48 hr, consistent with there being a reservoir of dendrimer conjugated AZD4320 effectively trapped for a long period of time at the tumour site. Retention in the tumour has been observed via imaging for similar dendritic delivery systems [21] and other nanoparticles [17].”
3.	Mathematical model. There is another assumption on the model that needs to be further discussed. That is assumption #3, “delivery of released AZD4320 from the plasma to the tumour without the dendrimer as a carrier is considered negligible”. This should be further justified. If not, then does this mean without the dendrimer as a carrier, the drug will not have an impact on the tumor? What about other free drugs in cancer treatment that have been tested in the past, or the free drugs that are currently in use in the clinic?	Thank you for this comment, and we understand this is somewhat unusual and it is specific/peculiar to this drug with its rather extreme physicochemical properties and rapid clearance. Indeed, unusually, the drug does not reach the tumour tissue without being conjugated to a dendrimer or another enabled complexing formulation such as a cyclodextrin. To make this clearer we have added text to the Discussion to address this comment, the additional text (highlighted in blue) is from line 537. This final assumption around AZD4320 delivery to tumour only being via the dendrimer conjugate is supported by preclinical studies which have shown that AZD4320 delivery to tumours, unusually, is highly dependent on formulation for efficacy. Formulations containing hydroxypropyl-β-cyclodextrin led to a significantly greater tumour exposure and efficacy when compared to alternative formulations. This is illustrated in Figure S2 where formulations of AZD4320 in 30% w/v hydroxypropyl-β-cyclodextrin and a micellar solubisation in 5% w/v Tween 80 are compared. Both formulations show similar plasma concentration time profiles but the Tween 80 formulation shows no efficacy and no drug tumour concentration. The AZD4320 drug itself, due to its very low solubility and very high protein binding does not

		distribute to the tumour without an enabled formulation. In the 30% hydroxypropyl-β-cyclodextrin formulation, the presence of cyclodextrin-drug complex leads to improved kinetics of exchange between plasma and the tumour. The combination of a high degree of both cyclodextrin and protein binding as well as the short half-life of the drug in plasma are thought to be responsible for this effect. Conjugating AZD4320 to a dendrimer both provides enhanced solubility and allows the dendrimer to act as a carrier for AZD4320 delivery to tumours. Therefore, delivery of AZD4320 to the tumour when released in the plasma from the dendrimer carrier is unlikely to happen, due to both its the low concentration in plasma as a result of the drug's rapid initial clearance and its physicochemical properties. Released drug access to the tumour is therefore assumed to be negligible in the model. The fact that the very fast releasing drug-dendrimer conjugate, SPL8933, gave lower concentrations in the tumour than the longer releasing drug-dendrimer conjugates also supports this assumption. This assumption would be not expected to hold for other drugs with more typical physicochemical properties or slower plasma clearance. In these cases, the model would also need to account for delivery of released drug in the plasma to the tumour as well as that delivered via the drug-dendrimer conjugate. This final assumption, also assumes that once the drug has reached the tumour it is retained there. This is supported by the fact that released concentrations of AZD4320 from SPL-8931 continue to increase up to 48 hr, consistent with there being a reservoir of dendrimer conjugated AZD4320 effectively trapped for a long period of time at the tumour site. Retention in the tumour has been also observed via imaging for similar dendritic delivery systems [21] and other nanoparticles [17].
4.	With respect to toxicity, only cardiovascular tolerability was analyzed and discussed. In fact, a much more appropriate approach is to use a whole-body PKPD model to analyze what impact(s) the drug or drug-dendrimer can have on other organs. It may be too late to do this now, and the reviewer understands that this model is a simplified version of a potentially very complex model. At least, the authors may want to address this in Discussion. The authors have cited a couple papers in this field already, e.g., this paper, Dogra et al.,	Thank you for this comment and these very helpful suggestions. We have addressed this comment in a number of ways and amended the manuscript. In this work, we focused on cardiovascular tolerability as our main toxicity endpoint. This is because cardiovascular toxicity issues were the limiting toxicities that had prevented the development of AZD4320 alone (and have raised issues in other Bcl-2/Bcl-x_l dual inhibitors). This is stated in the introduction (line 66). The objective of this model is to optimise the conjugate selection and design the nanomedicine in order to have an optimal release rate for this challenging molecule. The release rate is designed to minimise the systemic released drug plasma exposure, assumed to be responsible for many potential toxicities,

PMID: 30949850. Interestingly, this group has recently published a few more; worth having a look, PMID: 32314552, PMID: 32206211, PMID: 30382084, etc

without compromising the tumour exposure and thus efficacy and therefore to optimize this tumour Cmax to plasma Cmax ratio. It is tumour Cmax, or at least time over a particular concentration, which appears important for efficacy and the plasma Cmax which needs to be reduced to manage cardiovascular and other toxicities. Indeed thrombocytopenia is also a well-known class effect toxicity of dual Bcl-2/Bcl-x_L inhibitors and this is also improved when AZD4320 is delivered via drug- dendrimer conjugates of different release half-lives (Fig S7). Thrombocytopenia is shown to be manageable for AZD0466 in dog (Fig 7) . In this case, with this drug, we have managed to design the drug-dendrimer conjugate with this more simplified model and obtained good agreement with experimental data and thus enabled the progression of the candidate drug into the clinic.

The below text is already in the Discussion (line 691) and (line701). We have reviewed this text again and we feel that it explains our focus on cardiovascular toxicity and that by reducing plasma Cmax and concentrations of free drug in plasma we have improved other potential toxicities. We have also shown the thrombocytopenia data, a well-known on-target class effect for Bcl-2/x_L. In addition, more holistic toxicity is described as the no observed effect level in dog is reported which was determined by the cardiovascular toxicity, as the dose limiting toxicity, and this show was also observed to be well tolerated (see also Results section line 417). We also have added in the clinical trial identifier to enable a link to the clinical trial.

“As well as improving the cardiovascular tolerability of AZD4320, AZD0466 through weekly dosing was also found to have a reversible and dose-dependent decrease in platelet count (Fig. 7c). Platelet toxicity is one of the main on target toxicities Bcl-x_L inhibitors however dosing AZD4320 via the dendrimer conjugate shows the platelet decreases can be mitigated in rats (Fig. S7). In dogs, delivering AZD4320 via the dendrimer conjugate, AZD0466, shows only transient thrombocytopenia and that a suitable weekly dosing schedule allows for platelet recovery between doses (Fig. 7d). In addition, the no observed effect level for AZD0466 in dog was determined as 10 mg/kg (equivalent to ~3 mg/kg AZD4320) significantly higher than 0.2 mg/kg for AZD4320 itself (Table S5).”

In addition the Discussion ends with the sentence

“Delivering via a drug-dendrimer conjugate has enabled progression of an important dual Bcl-2/x_L inhibitor into clinical development. A full GLP toxicity package has been successfully completed in both rat and dog and an

investigational new drug (IND) application has been approved by the FDA and phase I clinical studies are in progress (NCT04214093).”

We recognise that the benefit of whole-body pharmacokinetic based models (e.g. PK-PD and PB-PK) models and that this approach would assess the impact on other organs as well as the key effect and toxicity organs and that this can be important for both nanomedicine design and understanding in their subsequent development. However, in this case, we were able to use a more simplified PK based model that allowed us to efficiently and effectively explore the critical factor of release rate on the efficacy and dose limiting toxicity and thus design a drug- dendrimer conjugate that has allowed us to a select development candidate and progress through full GLP toxicity studies and into clinical evaluation in patients.

We have addressed this comment our choice of approach, by discussing mathematical modelling in two places in the Discussion.

Firstly to provide some brief background of mathematical modelling in general to the nanomedicine field and its importance to aid the successful design and development of nanomedicines (line 496). This section also discusses why we have chosen to work with a more simplified model to efficiently design and select optimal release half-life for our drug-dendrimer conjugate. The whole text for this section is added in response to comment 1 above however, in reference specifically to question 4, we would like to highlight the below part of this additional text in the Discussion, line 508.

“To our knowledge, this is the first application of mathematical modelling, to obtain the desired release rate, a critical parameter in nanomedicine design and imperative to ensure optimal therapeutic index. Given the complexity of whole body PB-PK models for therapeutic nanomedicines where the drug needs to be released for activity, particularly during the design/discovery phase, the challenges of accurate bioanalysis of free drug from nanomedicines, in addition, the extreme physicochemical coupled with unusual pharmacokinetic properties of our drug, we elected to focus on developing a simple mathematical model to efficiently guide the design of our drug-dendrimer conjugate. This was focussed on predicting tumour concentrations, as the key effect organ, and minimising plasma concentrations as the surrogate for potential toxicities”

Secondly, to provide better context and discuss the mathematical modelling approaches more fully (as requested in point 1 as well) and to frame our work to the work of

		others using mathematical models in the field we have added to the Discussion, Line 654 and included these important references. The text is added in response to comment 1 above. We have look at activation cleaved caspase 3, a downstream marker of pathway activation and apoptosis, as PD maker in the tumour in this paper and there appears a good PK-PD correlation as shown in Figure 3b and correlation between cleaved caspase and tumour efficacy Figure 3. In other work, we have developed a PK-PD model to show that that intermittent targeting of Bcl-2 and Bcl-x_L via AZD4320, induces tumour regression yet manageable transient thrombocytopenia achieved through weekly scheduling (Balanchander et al, 2020, under revision, Clin Cancer Research). This model-based analysis showed that platelets appear to be exquisitely sensitive to Bcl-xL inhibition, compared to tumour cells, suggesting that on-target thrombocytopenia (at least of the transient kind) is an unavoidable feature of Bcl-xL activity. The complexity of the response to cleaved caspase in different tissues, this downstream marker of pathway activation, makes it challenging to develop a whole body PK-PD model based on this PD marker. This again supports our approach of using a PK based model to design our dendrimer conjugate.
5.	It is interesting to find that the authors have taken dose into account when attempting to define an overall optimization index where the predicted tumor C_{max} to plasma C_{max} ratio is effectively divided by the required dose. Would dosing frequency serve as another factor to be considered in this optimization index as well? At least, the authors may want to provide some discussions about this.	Thank you for this comment. We agree that, in addition to dose, we could have considered dosing frequency as well. The slower releasing linkers do lead to a better ratio of tumour C_{max} to plasma C_{max}, but the dose needs to be increased for the same efficacy which could lead to impractically large doses of drug-dendrimer conjugate (mentioned in the results section). More frequent dosing of the slower releasing constructs leading to a lower dose (per dosing event), would lead to a lower fraction of the dose reaching the target tumour and lower delivery efficiency. We therefore designed this to be the most efficient in terms of delivery efficiency. This has the important benefits of minimising cost of goods, polymer dose and better patient acceptability/convenience . We have added the following sentences in the Discussion to address this point, line 600. ..." or more frequent dosing which would also result in less efficient delivery (fraction of dose reaching the tumour). Efficiency of delivery is important from a cost of goods, minimising polymer dose to the patient and patient acceptability perspective in terms of number of vials and frequency of dosing."
6.	Figure 5.	1) Thank you for pointing out this errors, we have corrected 5c

1) Line 292: Fig. 5c should be Fig. 5e.
 2) Fig. 5f, where are the other two linker chemistries: SPL-8974 and SPL-8976?
 3) While the overall model performance in terms of predicting drug concentration behavior seems acceptable, in (g) it seems that the model only works for SPL-8974, not for SPL-8976 and SPL-8977; in (h), the model only works for the case of Released AZD4320

to 5e (line 303)

2) These two linkers are proprietary linkers, however we do not think it is essential to reveal the structures to understand the paper. We have however confirmed that they are designed to be susceptible to hydrolysis, like the other linkers used and have shared their in vitro release rate half-lives (Results section, line 322)

3) We agree that the model appears to work best for SPL-8974 in Figure 5g which shows the released data for all three of the conjugates in the second round of synthesis. The experimental data and model predictions for the released drug from SPL-8977 is the same in Figure 5g & Figure 5h; they are plotted on different scales as Figure 5h also shows total drug concentrations which are a lot higher. We feel the model does predict well for all three compounds. We have, however, gone back to the raw data and this does suggest a rogue point, for SPL-8976, as depicted below. We have therefore now omitted from experimental data which tightens the prediction for and data for this conjugate and have added a statement qualify in Figure legend "Data are means \pm SD (n=3)(except for SP-8976 where n=2 was used at final time point)".

The total AZD4320 concentration for SPL-8974, SPL 8976, SPL8977 and SPL8932 is now also shown in Figure S10 versus their predicted profile from the model support that this works for all the conjugates. This, again, together with Figure 5f supports the model for all 3 compounds made in this second round of synthesis. This figure also shows predicted vs experimental data for SPL8932; the compound we were trying to better on tolerability but wanted to maintain efficacy. In terms of model fit, it should be remembered there will be errors in the experimental data and that each data point is from a different mouse at each timepoint as this is a composite PK profile. In addition, the accurate bioanalysis of polymer conjugate/nanomedicines is notoriously complex (as

		discussed) particularly when extracting and quantifying released drug. Coupled with these challenges, the highly lipophilic nature of AZD4320 which presents additional analytical challenges, despite careful analysis some errors in the experimental data are to be expected. We have now added the correlation between predicted and experimental values for released drug concentration for all the 6 conjugates used in Figure S11 and this further supports the predictive power of the model. The following has been added to the text, in the Results section, line 327; ..”(Fig 5g) and total AZD4320 (Fig S10) was observed . The correlation between predicted and measured released AZD4320 tumour concentrations from all six drug-dendrimer conjugates is shown in Figure S11”. Additionally these additional figures confirming the predictive ability of the model are also referred to in the Discussion section, line 593. ...”Fig.5c & g, Fig. S10 & Fig. S11”
7.	Figure 6. Fig. 6b: data for 3mg/kg AZD4320 should be provided as well. Fig. 6c: data for 3mg/kg AZD4320 should be provided as well.	Thank you for pointing out this omission. We have updated both Fig. 6b and Fig. 6c updated with 10 mg/kg AZD0466 (equivalent to ~3mg/kg AZD4320) dos data

Two Figures in the main manuscript have been updated for clarity (Figure 1) and to include additional dose data as requested by Reviewer 2, point 7. These are shown below. We have also updated Figure 5g, as discussed to remove the rogue point and thus tighten the experimental data.

Fig. 1. Structure of initial AZD4320-dendrimer conjugates. Schematic of PEG₂₁₀₀ containing AZD4320-dendrimer conjugates showing (a) overall structure and (b) chemical structure of highlighted quadrant. Conjugation sites and synthetic route allow for a maximum of 32 conjugated PEG molecules and 32 AZD4320 molecules per dendrimer. X = CH₂ for SPL-8931, X = S for SPL-8932 and X = O for SPL-8933

Fig. 6. Efficacy of AZD0466 in RS4;11 tumour xenograft model (a) Tumour growth inhibition with AZD0466 treatment at various doses. Data in the graphs are represented as mean tumour volumes \pm SEM ($n = 6$ per group). (b) Kinetics of cleaved caspase 3 induction in tumours after a single administration of 10mg/kg, 34 mg/kg and 103 mg/kg AZD0466 doses. Data in the graphs are represented as mean \pm SD ($n = 3$ per timepoint per group) (c) Tumour concentration of total and released AZD4320 over time following a single dose of 10 mg/kg, 34 mg/kg or 103 mg/kg AZD0466. Data in the graphs are represented as mean \pm SD ($n = 3$ per timepoint per group).

REVIEWERS' COMMENTS:

Reviewer #1 (Remarks to the Author):

I believe the authors have answer and revised enough all my comments so the manuscript is ready for publication

Reviewer #2 (Remarks to the Author):

The authors have fully addressed my comments made to the first version of their paper. Thank you.